# IL-21/type I interferon interplay regulates neutrophil-dependent innate immune responses to *Staphylococcus aureus*

Rosanne Spolski[1,2]*, Erin E West[1,2], Peng Li[1,2], Sharon Veenbergen[1†],
Sunny Yung[3‡], Majid Kazemian[1,2§], Jangsuk Oh[1,2], Zu-Xi Yu[4],
Alexandra F Freeman[5], Stephen M Holland[5], Philip M Murphy[3],
Warren J Leonard[1,2]*

[1]Laboratory of Molecular Immunology, National Heart, Lung, and Blood Institute, National Institutes of Health, Bethesda, United States; [2]Immunology Center, National Heart, Lung, and Blood Institute, National Institutes of Health, Bethesda, United States; [3]Laboratory of Molecular Immunology, National Institute of Allergy and Infectious Diseases, National Institutes of Health, Bethesda, United States; [4]The Pathology Core, National Heart, Lung and Blood Institute, National Institutes of Health, Bethesda, United States; [5]Laboratory of Clinical Immunology and Microbiology, National Institute of Allergy and Infectious Diseases, National Institutes of Health, Bethesda, United States

*For correspondence:
spolskir@nhlbi.nih.gov (RS);
wjl@helix.nih.gov (WJL)

Present address: [†]Laboratory of Pediatric Gastroenterology, Erasmus University Medical Center, Rotterdam, The Netherlands; [‡]Sacremnto VA Medical Center, Internal Medicine/Infectious Diseases, Sacremnto, United States; [§]Departments of Biochemistry and Computer Science, Purdue University, West Lafayette, United States

**Competing interests:** The authors declare that no competing interests exist.

**Abstract** Methicillin-resistant *Staphylococcus aureus* (MRSA) is a major hospital- and community-acquired pathogen, but the mechanisms underlying host-defense to MRSA remain poorly understood. Here, we investigated the role of IL-21 in this process. When administered intra-tracheally into wild-type mice, IL-21 induced granzymes and augmented clearance of pulmonary MRSA but not when neutrophils were depleted or a granzyme B inhibitor was added. Correspondingly, IL-21 induced MRSA killing by human peripheral blood neutrophils. Unexpectedly, however, basal MRSA clearance was also enhanced when IL-21 signaling was blocked, both in *Il21r* KO mice and in wild-type mice injected with IL-21R-Fc fusion-protein. This correlated with increased type I interferon and an IFN-related gene signature, and indeed anti-IFNAR1 treatment diminished MRSA clearance in these animals. Moreover, we found that IFNβ induced granzyme B and promoted MRSA clearance in a granzyme B-dependent fashion. These results reveal an interplay between IL-21 and type I IFN in the innate immune response to MRSA.
DOI: https://doi.org/10.7554/eLife.45501.001

## Introduction

*Staphylococcus aureus* is a pathogenic bacterium responsible for a high percentage of hospital-acquired infections as well as for an increasing incidence of community-acquired skin and soft-tissue infections and secondary infections during pulmonary viral infections such as influenza (*Robinson et al., 2015*). The emergence of the highly virulent USA 300 lineage of strains of methicillin-resistant *S. aureus* (MRSA) has complicated efforts to control infections in immunocompromised patients as well as in healthy individuals (*Mediavilla et al., 2012*; *Rigby and DeLeo, 2012*). Neutrophils play a key role in host defense to pulmonary *S. aureus* infection and are critical for clearance of MRSA (*Parker and Prince, 2012a*; *Rigby and DeLeo, 2012*). Neutrophils are recruited to the lung and contribute to the immune response, along with proinflammatory cytokines, chemokines, and other anti-microbial mediators released by lung epithelial cells and resident and incoming innate immune cells (*Rigby and DeLeo, 2012*). This response can lead to tissue injury, compromising

pulmonary integrity and function, thereby promoting the pathogenesis of *S. aureus* pneumonia (*Parker et al., 2016*; *Parker and Prince, 2012a*).

Interleukin-21 was first identified as a T-cell derived cytokine with pleiotropic actions on lymphoid cells. Along with IL-2, IL-4, IL-7, IL-9, and IL-15, IL-21 shares the common cytokine receptor γ chain (*Asao et al., 2001*; *Leonard, 2001*), γc, which is mutated in humans with X-linked severe combined immunodeficiency (XSCID) (*Noguchi et al., 1993*), and defective IL-21 signaling substantially accounts for the defective B-cell function that is observed in XSCID patients (*Ozaki et al., 2002*; *Recher et al., 2011*), corresponding to IL-21's ability to promote terminal B cell differentiation to plasma cells (*Ozaki et al., 2004*). IL-21 also can promote the differentiation of Th17 cells (*Korn et al., 2007*; *Nurieva et al., 2007*; *Zhou et al., 2007*), is required for the normal development of T follicular helper cells (*Vogelzang et al., 2008*) and optimal Th2 responses to helminth infection (*Fröhlich et al., 2007*; *Pesce et al., 2006*), has potent activity as an anti-tumor agent (*Skak et al., 2008*; *Spolski and Leonard, 2014*), and promotes the development of a range of autoimmune diseases (*Bubier et al., 2009*; *Kwok et al., 2012*; *Spolski et al., 2008*), suggesting that blocking IL-21 may be a therapeutic approach in such diseases. Indeed, the effectiveness of JAK3 inhibitors in autoimmune disease presumably at least partially relates to the inhibition of IL-21 signaling.

IL-21 has also been shown to control both humoral and adaptive cellular responses to viral infections. For example, IL-21 is critical for the host response to chronic LCMV infection, as viral-specific CD8$^+$ T cell responses are impaired in *Il21r* KO mice (*Elsaesser et al., 2009*; *Fröhlich et al., 2009*; *Yi et al., 2009*). Moreover, although humoral immune responses to viral infection could initially develop in the absence of IL-21, the generation of long-lived plasma cell responses was defective in the absence of IL-21 (*Rasheed et al., 2013*). Previously, we found that IL-21 promotes the pathologic immune response to infection with Pneumonia Virus of Mice (PVM), which is highly related to human Respiratory Syncytial Virus (RSV), with enhanced neutrophil recruitment and acute respiratory distress (*Spolski et al., 2012*). In PVM, lung CD4$^+$ T cells constitutively express IL-21, and the number of IL-21$^+$ CD4$^+$ T cells increases following viral infection. Besides its effects on lymphoid cells, IL-21 exerts effects on non-lymphoid cell types, including inhibiting dendritic cell function and inducing the apoptosis of conventional DCs (*Brandt et al., 2003*; *Wan et al., 2013*); however, a role for IL-21 in innate neutrophil-mediated responses to pulmonary bacterial infection has not been reported. In light of the fact that human neutrophils have been reported to express IL-21R (*Takeda et al., 2014*), we explored the regulation of anti-bacterial responses by IL-21. Here, we investigated the ability of IL-21 to control a pulmonary model of MRSA infection.

## Results

### IL-21 enhances MRSA killing in the lungs of WT mice

To investigate the role for IL-21 in the response to a pulmonary bacterial infection, WT mice were intratracheally (i.t.) infected with the USA 300 strain of MRSA, which induced a significant increase in *Il21* mRNA (*Figure 1A*) and IL-21 protein (*Figure 1B*) in the lung 24 hr after infection. Moreover, using *Il21*-mCherry reporter transgenic (TG) mice, we found that lung CD4$^+$ T cells had low basal mCherry expression, but the percentage (*Figure 1—figure supplement 1, A and B*) and total number (*Figure 1—figure supplement 1C*) of *Il21*-mCherry expressing cells increased after infection with MRSA. IL-21 production was also significantly induced in CD4$^+$ T cells cultured in the presence of Staphylococcal enterotoxin B (SEB) (*Figure 1—figure supplement 1D*), suggesting that bacterial products can directly induce IL-21 production by this T cell population. Interestingly, treatment of WT mice with IL-21, as opposed to PBS, prior to intratracheal infection with MRSA resulted in a modest but significant increase in pulmonary clearance of bacteria at both 7 and 24 hr (*Figure 1C*; see summary of 6 independent experiments at 7 hr; *Figure 1—figure supplement 1E*). When we examined lung pathology, as expected, MRSA infection led to increased lung pathology compared to uninfected mice, but IL-21 treatment resulted in reduced lung pathology and neutrophil infiltration at 24 hr as compared to PBS-treated animals (*Figure 1D*, see insets; *Figure 1E*) at this time point. When we quantified neutrophils by flow cytometry, we found that IL-21-treated animals had more lung neutrophils than PBS-treated animals at 7 hr, but fewer lung neutrophils than PBS-treated mice at 24 hr (*Figure 1F*), consistent with the PBS vs. IL-21 effects on histology at this time point (*Figure 1D*). The specificity of the IL-21 effect was confirmed by the observation that IL-21 treatment

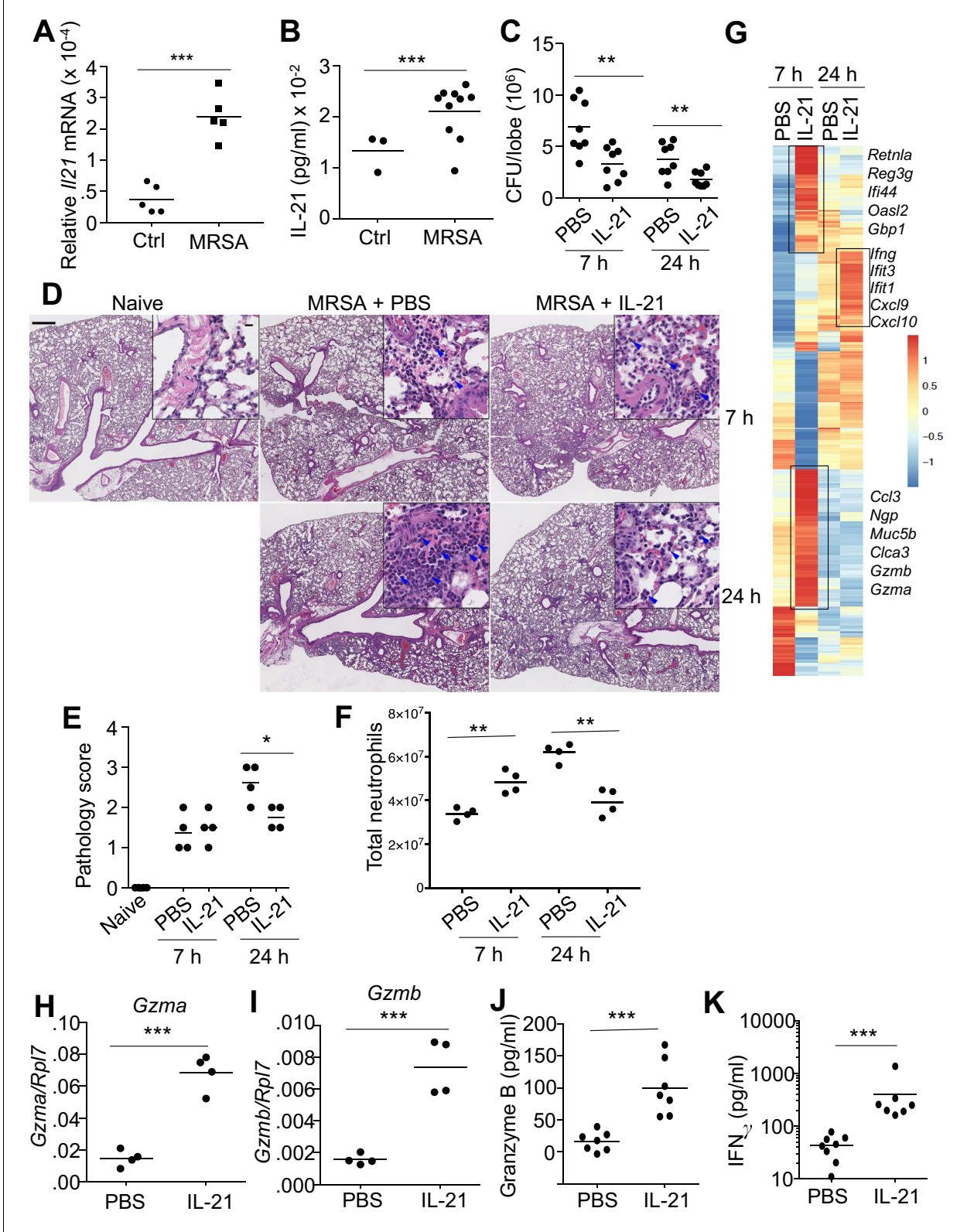

**Figure 1.** IL-21 enhances killing of pulmonary MRSA in WT mice. (A, B) Intra-tracheal infection with the USA 300 strain of MRSA induced a significant increase in pulmonary *Il21* mRNA (A) and IL-21 protein (B) 24 hr after infection. (C) PBS or 2 μg of IL-21 was administered i.t. to WT mice one day prior to MRSA infection, and lung MRSA CFU were quantitated 7 and 24 hr later. (D, E) Lung immunopathology was assessed in H and E-stained sections of lung tissue from naïve uninfected mice and mice pre-treated with PBS or IL-21 and then infected for 7 or 24 hr with MRSA (in upper left panel, the

*Figure 1 continued on next page*

Figure 1 continued

bar = 500 µm and inset bar = 10 µm) (D), and pathology (E) scores were assessed. (F–K) animals were infected with MRSA as above. (F) Total lung neutrophil cellularity was quantitated by flow cytometry after staining with Ly6G and CD11b. (G) RNA-Seq analysis was performed on total lung tissue mRNA (pools of 5 animals) isolated 7 or 24 hr after treatment of WT mice with PBS or IL-21. Boxed regions include genes mentioned in the text. (H–K) RT-PCR was used to assess expression of *Gzma* (H) and *Gzmb* (I) mRNA in lungs treated with IL-21 for 7 hr, and ELISA was used to assess the induction of granzyme B (J) and IFNγ protein (K) in corresponding bronchoalveolar lavage fluid at 7 hr. Data are representative of either three (A, B, C, F, H–K) or two (D, E, G) independent experiments and validation of RNA-Seq was performed by RT-PCR of mRNA from additional mice.

DOI: https://doi.org/10.7554/eLife.45501.002

The following figure supplements are available for figure 1:

**Figure supplement 1.** IL-21 is expressed in the lung before and after MRSA infection.

DOI: https://doi.org/10.7554/eLife.45501.003

**Figure supplement 2.** Effects of IL-21 on naïve lung.

DOI: https://doi.org/10.7554/eLife.45501.004

of *Il21r* KO mice did not significantly affect MRSA clearance (**Figure 1—figure supplement 1F**). Interestingly, treatment of naïve mice with IL-21 in the absence of MRSA infection induced an increased accumulation of cells in the BAL fluid (**Figure 1—figure supplement 2A**) and lung (**Figure 1—figure supplement 2B**), and a substantial percentage of the infiltrating cells were neutrophils (**Figure 1—figure supplement 2C**). This was accompanied by enhanced levels of the chemokines CXCL1 and MCP-1 (**Figure 1—figure supplement 2D and E**), both of which are involved in the recruitment of myeloid cells.

To investigate the mechanism by which IL-21 enhanced MRSA killing, we next performed RNA-Seq analysis using RNA from total lung tissue from mice that were treated with PBS or IL-21 prior to infection with MRSA. By 7 hr after infection of mice pre-treated with IL-21, we observed differential expression of a range of genes, including those encoding the cytolytic proteases granzyme A (*Gzma*) and granzyme B (*Gzmb*) as well as genes involved in chemoattraction (*Ccl3*), mucus hyper-production (*Muc5b, Clca3*), and anti-microbial defense (e.g., *Retnla, Reg3g, Ifi44, Oasl2, Gbp1,* and *Ngp*) (**Figure 1G** and **Supplementary file 1**) (**Dietert et al., 2014**; **Zanin et al., 2016**). At 24 hr after MRSA infection, lungs from IL-21-pre-treated mice expressed *Ifng* mRNA and interferon-induced mRNAs (*Ifit1, Ifit3*) as well as transcripts for major interferon-inducible chemokines involved in recruitment of myeloid and lymphoid cells (*Cxcl9, Cxcl10*) (**Figure 1G**). We used RT-PCR to confirm IL-21-induced expression of *Gzma* and *Gzmb* mRNAs in the lung at 7 hr after infection (**Figure 1H and I**), and IL-21-induced granzyme B and IFNγ protein production was detected by ELISA in bronchoalveolar lavage fluid at 7 hr after infection (**Figure 1J and K**).

## IL-21 induces granzyme-mediated MRSA killing by neutrophils

Because the analysis of lung pathology revealed a significant infiltration of neutrophils early after MRSA infection as compared to uninfected mice (**Figure 1D and F**), we next investigated the specific response of neutrophils to IL-21. Previously, human peripheral neutrophils were reported to either not express (**Pelletier et al., 2004**) or to express IL-21R (**Takeda et al., 2014**). Consistent with the latter report, mouse bone marrow neutrophils (Ly6G⁺CD11b⁺) expressed low levels of surface IL-21R as assessed by flow cytometry. Interestingly, IL-21R expression was enhanced by treatment with peptidoglycan (**Figure 2A** and **Figure 2—figure supplement 1A**), a component of the *S. aureus* cell wall. Thus, *S. aureus* components can induce expression of the IL-21 receptor (**Figure 2A**) as well as the IL-21 ligand (**Figure 1A and B**). To determine whether IL-21-induced clearance of MRSA required neutrophils, WT mice were treated with anti-Ly6G to deplete neutrophils (**Figure 2B**) and then given PBS or IL-21 i.t. prior to MRSA infection. Although IL-21 treatment significantly reduced the MRSA titer in the lung 24 hr after infection of isotype control-treated mice, it had less of an effect on MRSA clearance in the neutrophil-depleted mice (**Figure 2C**). Consistent with a major role for neutrophils in MRSA clearance, neutrophil-depleted lungs had significantly higher CFU than lungs not subjected to neutrophil depletion, including in the PBS-treated animals (**Figure 2C**). The remaining smaller effect of IL-21 on MRSA clearance in the absence of neutrophils might be due to effects of IL-21 on phagocytosis by macrophages (**Vallières and Girard, 2013**) or dendritic cell survival (**Wan et al., 2013**). Although IL-21 enhanced *Gzma* and *Gzmb* mRNA expression in isotype control treated mice, it no longer did so when neutrophils were depleted by anti-Ly6G treatment,

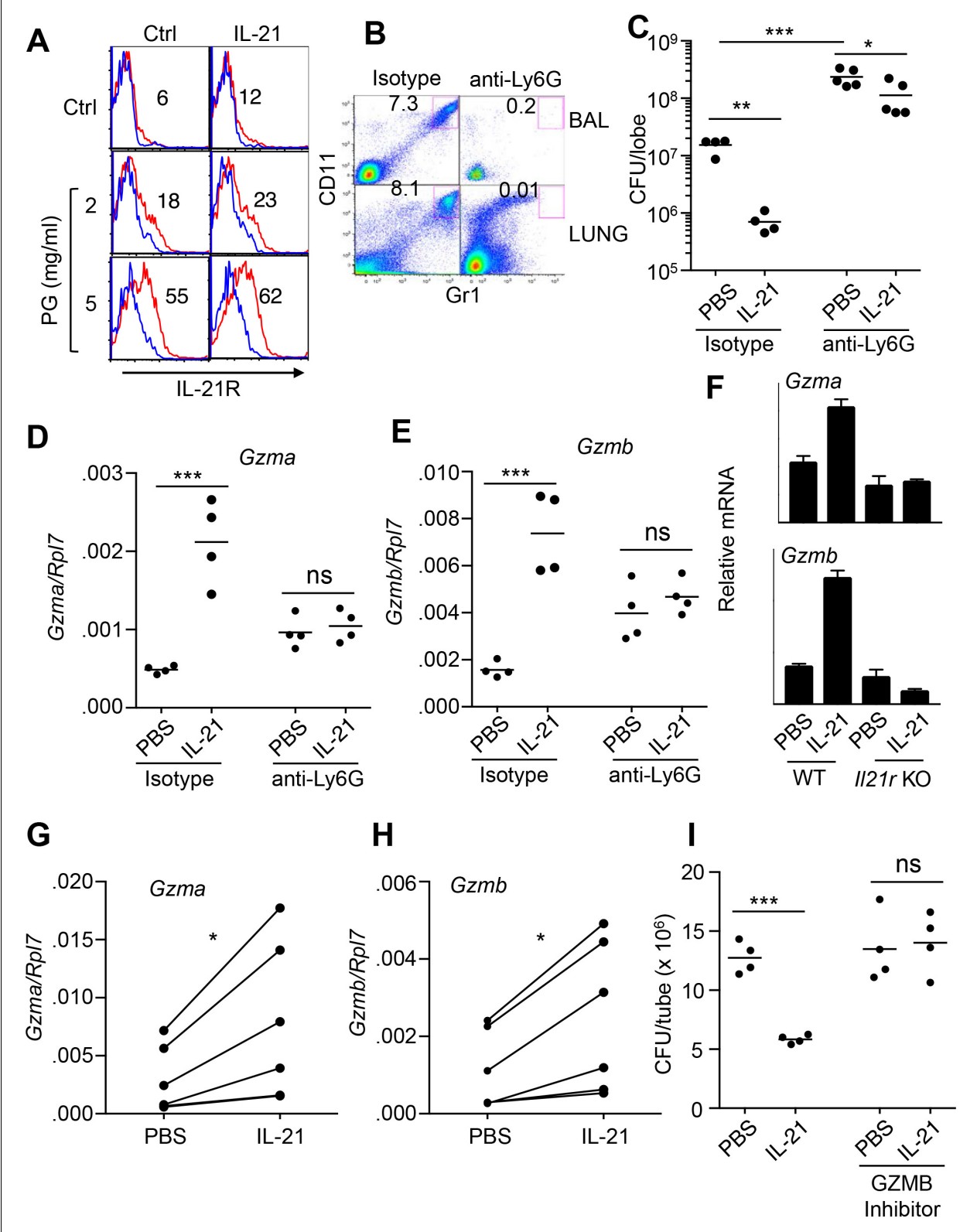

**Figure 2.** IL-21-mediated granzyme production and MRSA killing by lung neutrophils. (**A**) Bone-marrow neutrophils were purified by negative selection using a neutrophil isolation kit (Miltenyi) and either not stimulated or stimulated in vitro with IL-21 (100 ng/ml) for 24 hr in the absence (control) or presence of peptidoglycan (PG) (2 or 5 μg/ml); IL-21R expression on gated Ly6G⁺CD11b⁺ neutrophils was detected by flow cytometry (blue = isotype control; red = anti-IL-21R). (**B, C**) IL-21 lowers the MRSA CFU in a neutrophil-dependent fashion. Mice were treated i.p. with an isotype control mAb or

*Figure 2 continued on next page*

Figure 2 continued

neutrophil-depleted with anti-Ly6G (1A8 mAb), and the efficacy of neutrophil depletion is shown in BAL and lung (B). Mice were then treated with PBS or IL-21 i.t., infected i.t. with MRSA, and CFU quantitated at 24 hr (C). CFU quantitation was performed on the left single lobe, whereas immune populations and RNA were determined using the right lobes. (D, E) Levels of *Gzmb* and *Gzma* mRNAs in lung tissue of untreated or neutrophil-depleted mice (treated as in panel C) were quantitated by RT-PCR. (F) Levels of *Gzma* and *Gzmb* mRNAs in purified cell-sorted lung neutrophils (>98% pure Ly6G$^+$CD11b$^+$) from either untreated or IL-21-treated WT or *Il21r* KO mice were measured by RT-PCR and normalized to *Rpl7* expression. (G, H) Lung neutrophils were elicited by i.t treatment with heat-killed *S. aureus* 24 hr prior to isolation, purified, stimulated in vitro for 4 hr with either PBS or IL-21 (100 ng/ml), and *Gzma* (G) and *Gzmb* (H) mRNAs assayed by RT-PCR and normalized to *Rpl7* expression. (I) Purified HKSA-elicited lung neutrophils were incubated in vitro with MRSA for 3 hr in the presence of PBS or IL-21 either without or with the granzyme B inhibitor, Z-AAD-CMK. MRSA CFU was quantitated by plating serial dilutions on blood agar plates. Representative experiments are shown; each experiment was performed three times with similar results.

DOI: https://doi.org/10.7554/eLife.45501.005

The following figure supplement is available for figure 2:

**Figure supplement 1.** Purity of neutrophil preparations.
DOI: https://doi.org/10.7554/eLife.45501.006

implicating neutrophils as a major source of these IL-21-induced transcripts in MRSA-infected mice (*Figure 2D and E*). Basal *Gzma* and *Gzmb* mRNA levels were higher in lung tissue in animals treated with anti-Ly6G as compared to isotype control antibody; the basis for the apparent increase in these transcripts after neutrophil depletion is unclear.

To confirm that *Gzma* and *Gzmb* mRNAs were specifically induced by IL-21's interaction with its receptor on lung neutrophils, WT and *Il21r* KO mice were treated with IL-21 for 7 hr, and highly purified (>98%) FACS-sorted lung neutrophils were analyzed by RT-PCR; indeed, *Gzma* and *Gzmb* mRNAs were induced by IL-21 in neutrophils from WT but not *Il21r* KO mice (*Figure 2F*). To further assess whether granzymes were involved in IL-21-mediated clearance of MRSA, WT mice were primed intratracheally with heat-killed *S. aureus* (HKSA), and recruited lung neutrophils were purified 24 hr later (*Figure 2—figure supplement 1B*). IL-21 could induce expression of both *Gzma* and *Gzmb* mRNA (*Figure 2G and H*) and augmented MRSA killing by these cells in vitro (*Figure 2I*), whereas this IL-21-induced killing was prevented when a granzyme B inhibitor, Z-AAD-CMK, was added (*Figure 2I*).

## IL-21 also induces a cytolytic profile in human neutrophils

Analogous to the effects of IL-21 on mouse neutrophils, highly purified (>99%) human peripheral blood neutrophils that showed no evidence of contamination by NK or CD8 T cells (*Figure 3—figure supplement 1A*) expressed *IL21R* mRNA and expression was further induced by stimulation with IL-21 (*Figure 3A*), and we furthermore found that surface IL-21R protein expression was modestly enhanced by stimulation with IL-21 (*Figure 3B*, left and right panels). Moreover, when we performed RNA-Seq analysis on highly purified human neutrophils that were either not stimulated or stimulated with IL-21, consistent with our observations in IL-21 treated mice, *GZMA*, *GZMB*, *GNLY* (encoding granulysin), *PRF1* (encoding perforin) mRNAs were all induced more than 2-fold by IL-21, and this was observed whether or not cells were exposed to heat-killed *S. aureus* (HKSA) (*Figure 3C* and *Supplementary file 2*). We confirmed the induction of *GZMA*, *GZMB*, and *GNLY* by RT-PCR (*Figure 3D-F*). Moreover, although *IFNG* mRNA induction did not quite meet the FC >2 criterion and thus was not included in *Figure 3C*, expression of this gene was highly induced by IL-21, as assessed by RT-PCR (*Figure 3G*). Interestingly, IL-21 treatment did not appreciably alter the expression of intracellular granzyme B protein detected by flow cytometry in permeabilized cells (*Figure 3H*, left and right panels), but it significantly increased the release of granzyme B protein into the supernatant of purified human peripheral blood neutrophils, as assessed by ELISA (*Figure 3I*). To further analyze production of granzyme B by neutrophils, we used western blotting. Levels of granzyme B from >99% enriched neutrophils were lower than in an equivalent number of PBMC (*Figure 3—figure supplement 1B*), but PBMC contamination could not account for the signal from purified neutrophils, as even a 1% level of PBMCs gave no detectable signal when added to A549 cells, which do not express detectable Granzyme B, and even a 5% PBMC addition was still less than the signal from the >99% enriched neutrophils (*Figure 3—figure supplement 1B*). Coupled to flow cytometry (*Figure 3H*) and the failure of IL-21 to induce *Gzmb* mRNA after neutrophil

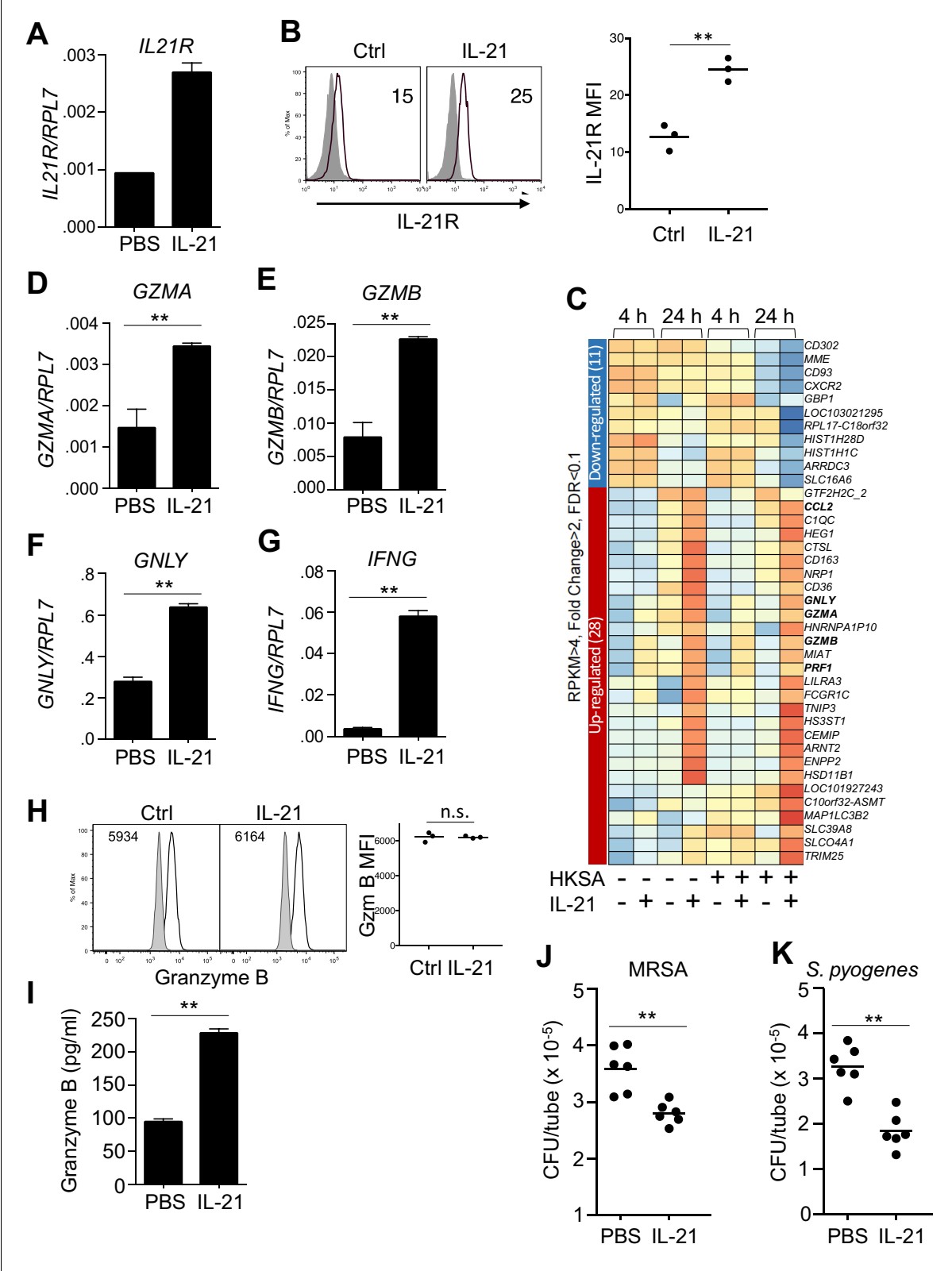

**Figure 3.** Human neutrophils express IL-21R, and IL-21 induces a program that leads to granzyme B-dependent neutrophil-mediated killing of MRSA. (A, B) Purified peripheral blood neutrophils were stimulated with PBS (control) or IL-21 (100 ng/ml) for 4 hr, and *IL21R* mRNA was measured by real-time PCR and normalized to *RPL7* expression (A), and IL-21R protein levels were measured by flow cytometry (B); left panel shows isotype control shaded and anti-IL21R black line, with a summary in the right panel. (C) RNA-Seq was performed on neutrophils after 4 or 24 hr incubation with PBS or IL-21 in

*Figure 3 continued on next page*

*Figure 3 continued*

the absence or presence of heat-killed *S. aureus* (10⁶/ml). We used HKSA rather than live bacteria in order to allow analysis at 24 hr, as live bacteria would have overgrown the system by then. Genes differentially expressed (fold-change >2.0) are shown. Shown is a representative RNA-Seq analysis. (D – G) Human peripheral blood neutrophils were stimulated for 4 hr in vitro in the presence or absence of IL-21 and *GZMA* (D), *GZMB* (E), *GNLY* (F), and *IFNG* (G) mRNA levels were quantitated by RT-PCR and normalized to *RPL7* expression. (H) Purified human neutrophils were stimulated with IL-21 for 4 hr, fixed, permealized, and stained for intracellular granzyme B protein (gated on CD66b⁺ cells); MFIs are summarized in right panel. (I) Granzyme B protein was measured by ELISA in supernatants from human peripheral blood neutrophils cultured for 24 hr in either the absence or presence of IL-21 (100 ng/ml). (J, K) Human neutrophils were incubated in vitro with either MRSA (J) or *S. pyogenes* (K) for 3 hr with PBS or IL-21 (100 ng/ml). MRSA and *S. pyogenes* CFU were quantitated by plating serial dilutions on blood agar plates. Results shown are representative of 3 independent experiments, except panel C shows one of two similar independent RNA-Seq experiments, each from a different donor.
DOI: https://doi.org/10.7554/eLife.45501.007

The following figure supplements are available for figure 3:

**Figure supplement 1.** Purity of human neutrophils.
DOI: https://doi.org/10.7554/eLife.45501.008

**Figure supplement 2.** Reactive oxygen species were measured by flow cytometry in CellRox Red loaded peripheral blood neutrophils that were stimulated at 37°C for 30 min with PBS, IL-21, HKSA, or HKSA +IL-21.
DOI: https://doi.org/10.7554/eLife.45501.009

depletion of mice (*Figure 2E*), this further supports production of granzyme B by these cells (see Discussion). A previous report had shown enhanced reactive oxygen species (ROS) in response to zymosan plus IL-21 (*Takeda et al., 2014*). However, IL-21, either alone or in combination with HKSA, did not significantly enhance intracellular ROS production (*Figure 3—figure supplement 2*), suggesting that increasing ROS is not a major mechanism by which IL-21 augmented the killing of MRSA. Although granzymes A and B were originally characterized based on their roles in T cell and NK cell-mediated killing of target cells (*Voskoboinik et al., 2015*), these proteases and anti-microbial peptides can promote the killing of a number of types of bacteria as well (*Walch et al., 2014*). IL-21 also induced expression of the *CCL2* gene (*Figure 3C*), which encodes MCP1, a chemokine that recruits additional innate cells to sites of infection (*Balamayooran et al., 2011*). Thus, IL-21 induces a transcriptional program in human neutrophils that enhances their cytotoxic potential and leads to further recruitment of innate immune cells. Because IL-21 enhanced release of granzyme B, we investigated if IL-21 stimulation could enhance in vitro killing of MRSA by these cells. Indeed, purified peripheral human blood neutrophils stimulated with IL-21 consistently reduced the number of MRSA (*Figure 3J*). Moreover, IL-21 also induced clearance of *Streptococcus pyogenes* by human neutrophils (*Figure 3K*), indicating that the effect was not restricted to a single pathogen.

## Unexpected enhanced MRSA clearance and lung inflammation in *Il21r* KO mice

Because IL-21 augmented MRSA clearance, we hypothesized that *Il21r* KO mice, as compared to WT mice, would exhibit defective bacterial killing and thus a greater CFU when infected with MRSA. Unexpectedly, however, when mice were infected i.t. with MRSA, although no difference was observed at 4 hr, the bacterial burden in the lungs of *Il21r* KO mice at 24 hr was significantly lower than that observed in WT mice (*Figure 4A*), indicating greater clearance of the bacteria. To confirm this finding, we next directly compared in a single experiment the levels of MRSA killing for WT versus *Il21r* KO mice and for WT mice treated with PBS versus IL-21, and indeed, as compared to untreated WT mice, *Il21r* KO mice and IL-21-treated WT mice both had lower MRSA CFUs (*Figure 4B*). To further evaluate the ability of WT vs. *Il21r* KO mice to handle MRSA, we examined the inflammatory response. In WT mice, infection with Pneumonia Virus of Mice (PVM), in which host defense substantially depends on adaptive immunity, results in a marked inflammatory response by 5 or 6 days, and we previously showed that this is substantially diminished after infection of *Il21r* KO mice (*Spolski et al., 2012*). In contrast, with MRSA infection, in which host defense significantly depends on a neutrophil-mediated innate immune response, *Il21r* KO mice had greater inflammation than WT mice, and this was already evident at 4 and 24 hr (*Figure 4C*, see magnified insets), with an increased pathology score at both time points (*Figure 4D*). Interestingly, when infiltrating neutrophils were isolated and counted, both IL-21-treated WT mice (see *Figure 1F*) and *Il21r* KO mice (*Figure 4E*) had more neutrophils than untreated WT mice at the early time point but fewer

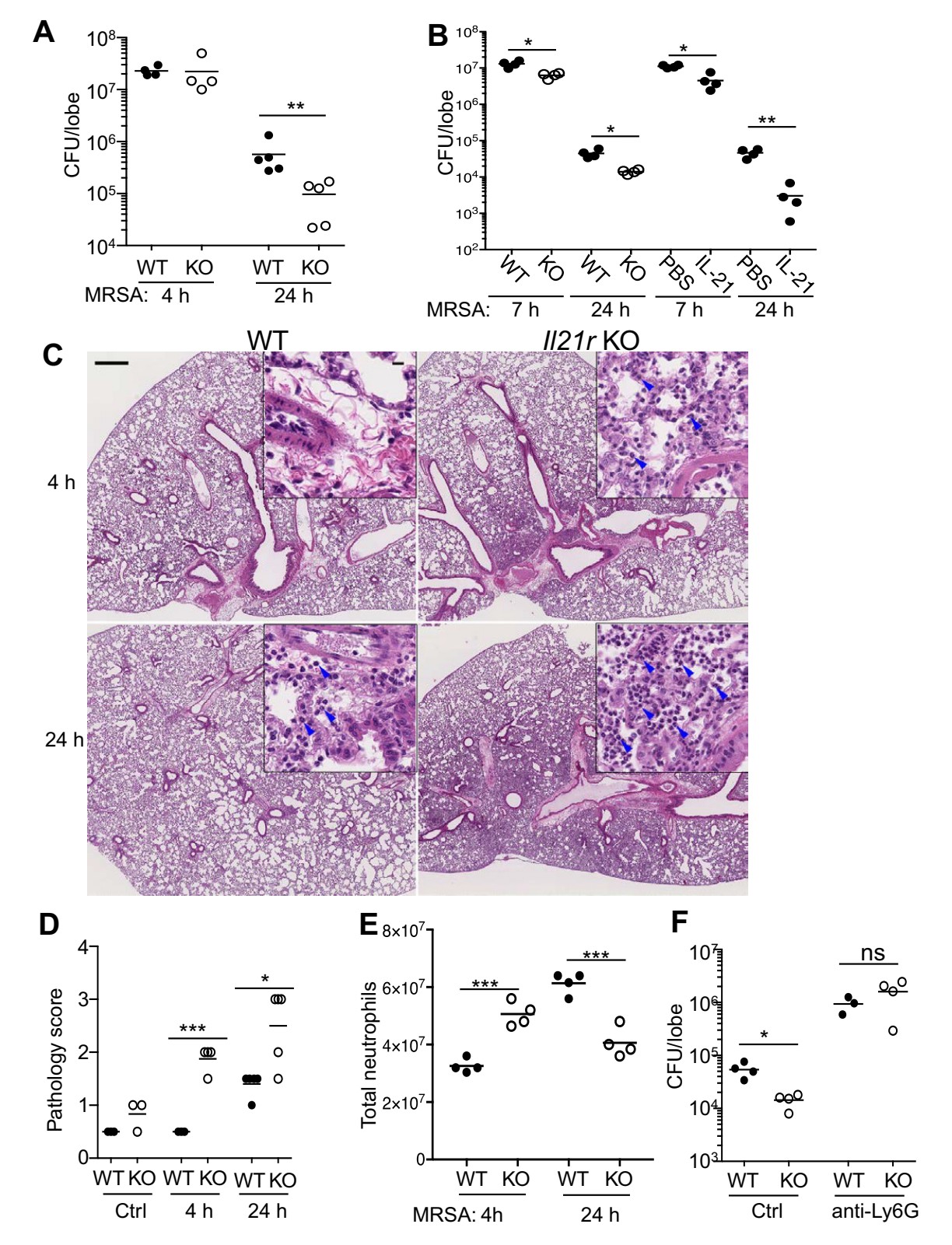

**Figure 4.** Pulmonary MRSA infection is cleared more efficiently in *Il21r* KO than in WT mice and this requires neutrophils. (**A**) WT and *Il21r* KO mice were infected intratracheally with MRSA ($2 \times 10^7$) and CFU in the lung were quantitated at 4 and 24 hr post-infection. (**B**) WT and *Il21r* KO mice were infected in parallel with WT mice that had been treated with PBS or 2 μg IL-21, and CFU in the lung were quantitated at 7 and 24 hr post infection. (**C–E**) Lung immunopathology was assessed in H and E-stained lung sections (bar in upper left panel = 500 μm; bar in inset = 10 μm) (**C**); lung neutrophil
*Figure 4 continued on next page*

*Figure 4 continued*

cellularity as assessed by pathology score (D) and flow cytometry (E) were assessed at 4 and 24 hr post-infection. (F) WT or *Il21r* KO mice were pre-treated with an isotype control antibody or anti-Ly6G to deplete neutrophils, infected with MRSA, and CFU quantitated at 24 hr. Representative experiments are shown; each experiment was performed three times with similar results.

DOI: https://doi.org/10.7554/eLife.45501.010

neutrophils than untreated WT mice at 24 hr; this is potentially reflective of the better clearance of MRSA in the *Il21r* KO mice by 24 hr (*Figure 4A*). Note that the higher pathology score at 24 hr (*Figure 4D*) is not inconsistent with the lower total neutrophil count at that time point (*Figure 4E*), as the pathology score is not limited to neutrophil numbers but rather to the severity of lung lesions in involved areas. WT lungs at 24 hr showed only focal lesions, mostly seen at perivascular areas with mild inflammation. These inflammatory cells consist of neutrophils, lymphocytes and macrophages. However, KO lung at 24 hr showed diffuse lesions with alveolitis and numerous neutrophils, lymphocytes, and macrophages. Neutrophils were important for the enhanced anti-bacterial activity observed in *Il21r* KO mice, as neutrophil depletion prevented the enhanced clearance of MRSA (*Figure 4F*).

## Type I interferons contribute to the enhanced MRSA clearance by *Il21r* KO mice

To investigate the molecular mechanism(s) underlying the enhanced MRSA clearance in *Il21r* KO mice, we used RNA-Seq to compare gene expression patterns in WT versus *Il21r* KO lungs at 4 and 24 hr after MRSA infection (*Figure 5A* and *Supplementary file 3*). Of the 1474 differentially-expressed genes, 664 were downregulated and 810 were upregulated in the *Il21r* KO mice (*Figure 5A*). Some of the genes that were most increased in the *Il21r* KO lung at both time points are known to be regulated by type I and type II IFNs, including multiple members of the *Gbp* (*Gbp1, Gbp2, Gbp3, Gbp4, Gbp6, Gbp8, Gbp10*) and *Ifi* (*Ifi44, Ifi45, Ifit1, Ifit2, Ifit3*) families (*Figure 5B* and *Supplementary file 3*), which mediate responses to bacterial and viral infections (*Berry et al., 2010*; *Yamamoto et al., 2012*). Analysis of the top five upstream regulators of the most differentially expressed genes identified the type I IFN receptor (IFNAR1) and IFNγ as being key pathways in the differential regulation of transcripts in *Il21r* KO versus WT cells (*Figure 5C*). Moreover, the IFNλ receptor (IFNLR1) and miR-21, a target of type 1 and 3 IFNs (*Liu et al., 2017*; *Yang et al., 2010*), were also identified (*Figure 5C*). To determine whether the local production of type I IFNs might account for the gene expression profile seen in the RNA-Seq analysis, we measured IFNα protein levels by ELISA in bronchoalveoar lavage (BAL) fluid at 4 hr after MRSA infection, and levels were higher in the *Il21r* KO than in the WT samples (*Figure 5D*). The higher IFNα in the *Il21r* KO BAL correlated with better MRSA clearance in *Il21r* KO than in WT mice (lower CFU); however, blocking with anti-IFNAR1 raised the CFU for both WT and KO mice to similar levels (*Figure 5E*), supporting the idea that an augmented type I IFN response in the *Il21r* KO mice (*Figure 5D*) indeed was responsible for their enhanced MRSA killing (*Figure 5E*, left). Pre-treatment with anti-IFNAR1 had only a modest effect on lung cellularity (not achieving statistical significance) (*Figure 5F*). Moreover, besides increasing the CFU for WT and *Il21r* KO mice to a similar level (*Figure 5E*), anti-IFNAR1 treatment also blocked the enhanced expression of interferon-regulated genes, *Gbp1* and *Oasl1*, in lungs from *Il21r* KO mice (*Figure 5G and H*).

## Blocking pulmonary IL-21 signaling phenocopies the enhanced antimicrobial activity of *Il21r* KO mice

It was possible that the enhanced MRSA killing in the *Il21r* KO mice was a developmentally-acquired phenotype rather than directly resulting from defective IL-21 signaling. We therefore intratracheally administered IL-21R-Fc fusion protein into WT mice for two days prior to MRSA infection to acutely block IL-21R-dependent signaling. Analogous to the *Il21r* KO mice, WT mice receiving IL-21R-Fc had greater clearance of MRSA at both 4 and 24 hr after infection than was observed in mice treated with the control Fc (*Figure 6A*). Moreover, histological analysis of lung sections showed that mice treated with IL-21R-Fc had enhanced inflammation (*Figure 6B*) and significantly increased pathology and neutrophil scores by 4 hr after infection (*Figure 6—figure supplement 1, A and B*), analogous

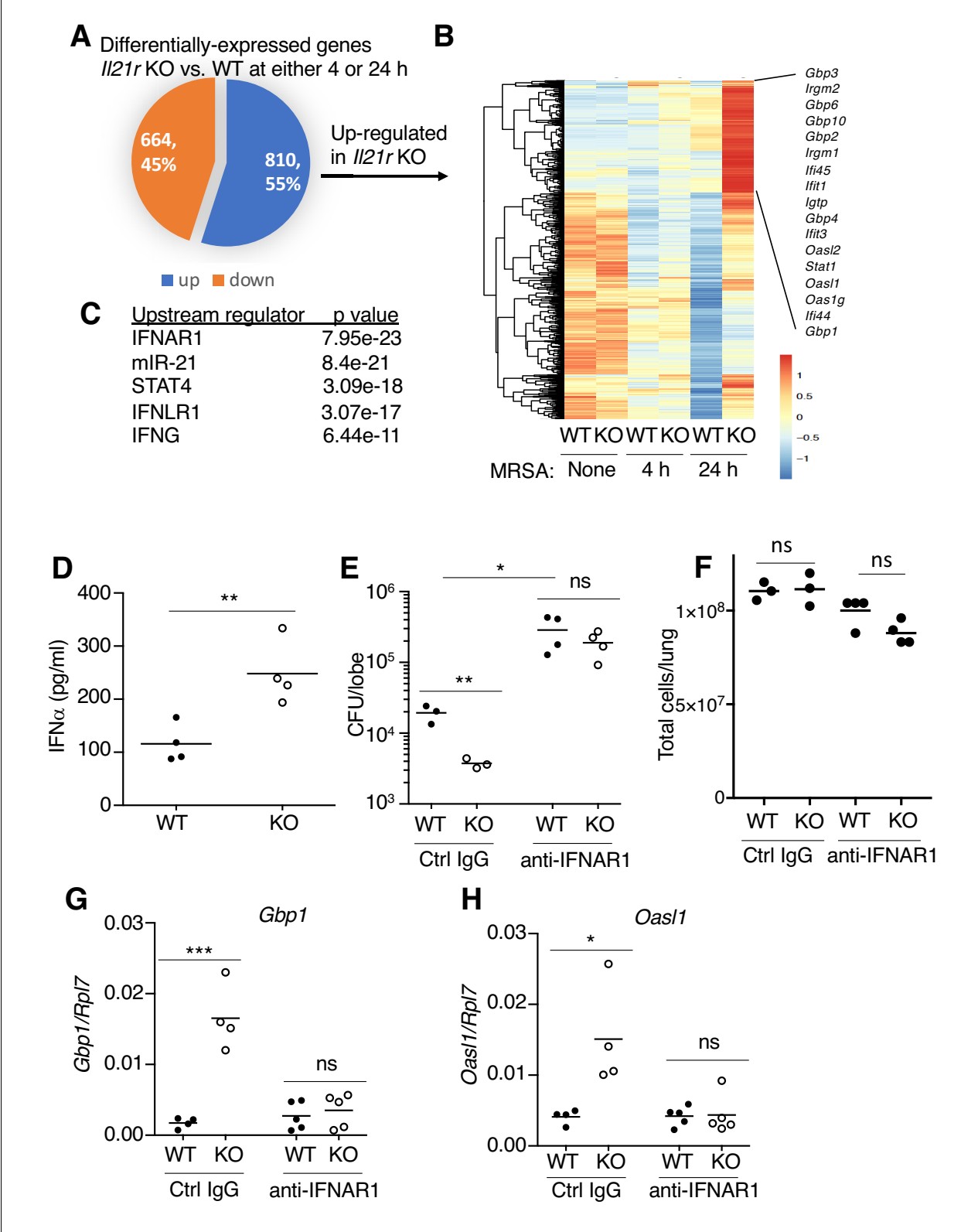

**Figure 5.** RNA-Seq analysis revealed an enhanced IFN profile in *Il21r* KO lungs. (**A**) Venn diagram showing the number of differentially expressed genes (1474) in WT vs *Il21r* KO lungs, with 664 genes downregulated and 810 genes upregulated in the *Il21r* KO. (**B**) Heat map showing a cluster of IFN-related genes more highly expressed in *Il21r* KO than in WT lung. (**C**) Pathway analysis of differentially expressed genes shows enrichment for IFN-related genes. (**D**) IFNα was measured by ELISA in BAL fluid from WT and *Il21r* KO mice 4 hr after infection with MRSA. (**E**) Diminished MRSA killing in

*Figure 5 continued on next page*

Figure 5 continued

lungs at 24 hr in mice pre-treated with anti-IFNAR1. (F) Total lung cellularity at 24 hr in mice pre-treated with Ctrl IgG or anti-IFNAR; the differences observed were not statistically significant at p<0.05. (G, H) Blocking type I IFN signaling with anti-IFNAR1 prior to infection with MRSA prevented enhanced expression of *Gbp1* (G) and *Oasl1* (H) mRNAs at 24 hr in *Il21r* KO lungs. Expression was normalized to *Rpl7*, Representative results from one of 2 independent RNA-Seq experiments are shown in panel A-C; in D-H, each experiment was performed three times with similar results.
DOI: https://doi.org/10.7554/eLife.45501.011

to what we observed above in the *Il21r* KO mice (*Figure 4D and E*). Moreover, increased IFNα protein was detected at 4 hr in the BAL fluid of mice receiving IL-21R-Fc (*Figure 6C*), analogous to the increased IFNα we had detected in the BAL fluid of *Il21r* KO mice (*Figure 5D*). Consistent with increased type I interferon production, RNA-Seq analysis of mRNA from lungs of MRSA-infected mice that were treated with the IL-21R-Fc fusion protein revealed augmented expression of mRNAs for a range of genes regulated by type I or type II interferons, as compared to mice treated with the control Fc (*Figure 6—figure supplement 1C* and *Supplementary file 4*). Thus, the absence of IL-21 signaling, as occurred in lungs from both *Il21r* KO mice and WT mice treated with IL-21R-Fc, led to an augmented interferon-responsive gene signature and enhanced MRSA killing.

The fact that there was enhanced IL-21-induced granzyme-mediated MRSA killing in WT mice and yet also enhanced killing in the absence of IL-21 signaling prompted us to further examine how the increased levels of type I interferon might explain the enhanced MRSA killing in the *Il21r* KO mice and in WT mice treated with the IL-21R-Fc fusion protein. Interestingly, at 24 hr after MRSA infection, the *Il21r* KO mice had higher levels of *Gzmb* mRNA in lung tissue compared to WT mice, and treatment of mice with an anti-IFNAR1 blocking antibody reduced *Gzmb* mRNA in the *Il21r* KO lungs to WT levels (*Figure 6D*). WT mice pre-treated with IL-21R-Fc prior to MRSA infection also had elevated levels of lung *Gzmb* mRNA at both 4 and 24 hr (*Figure 6E*). Moreover, IFNβ not only enhanced *GZMB* mRNA expression by purified normal human neutrophils to a similar degree as did IL-21 (*Figure 6F*), but it also induced killing of MRSA in a granzyme-dependent manner given that a significant decrease in CFU was no longer observed when a GZMB inhibitor was added (*Figure 6G*). Thus, both IL-21 and type I interferon appear to induce similar cytotoxic programs and to augment killing of MRSA.

The enhanced type I interferon responses that we observed when IL-21 signaling was absent or impaired suggested an inverse correlation between IL-21 and IFNα/β levels during the response to MRSA infection. We explored this possibility using an in vitro system in which both cytokines could be produced. Specifically, we examined the response of CD4$^+$ T cells and bone marrow-derived dendritic cells in a mixed lymphocyte response (MLR) to the SEB superantigen. Although the addition of SEB with control Fc induced *Il21* mRNA production by CD4$^+$ T cells in this MLR (*Figure 6H*), the addition of IL-21R-Fc augmented the effect. Interestingly, dendritic cell production of type I IFN mRNAs (*Ifna2* and *Ifnb*) was inhibited by SEB (*Figure 6I and J*); importantly, IL-21 is at least partially responsible for inhibiting type I IFN production, as addition of IL-21R-Fc fusion protein instead of control Fc diminished or reversed the inhibition (*Figure 6I and J*). It has previously been demonstrated that IFNα in combination with either IL-12 or TCR activation can enhance expression of IL-21 in human NK and T cells and downregulate IL-21R expression on these cells (*Strengell et al., 2004*), in keeping with the functional interaction of these two signaling pathways.

## IL-21 does not enhance MRSA killing by neutrophils from AD-HIES patients

To further investigate the mechanism of IL-21-induced killing of MRSA, because IL-21 signals in part via STAT3, we next used neutrophils from patients with autosomal dominant hyper-IgE syndrome (AD-HIES) or Job's syndrome, a disease caused by autosomal dominant mutations in *STAT3* that result in STAT3 deficiency, with increased susceptibility to fungal and bacterial infections, including *S. aureus* (*Freeman and Holland, 2008*). When peripheral blood neutrophils from three healthy donors and six patients were assayed in a MRSA killing assay, IL-21 significantly lowered the CFU with cells from all normal donors (ND1, ND2, ND3), but it did not significantly lower the CFU in experiments using cells from any of the AD-HIES patients (PT1-PT6; *Figure 7A*). In fact, two of the six patients (PT3 and PT4) had higher basal MRSA killing than did the corresponding normal donor,

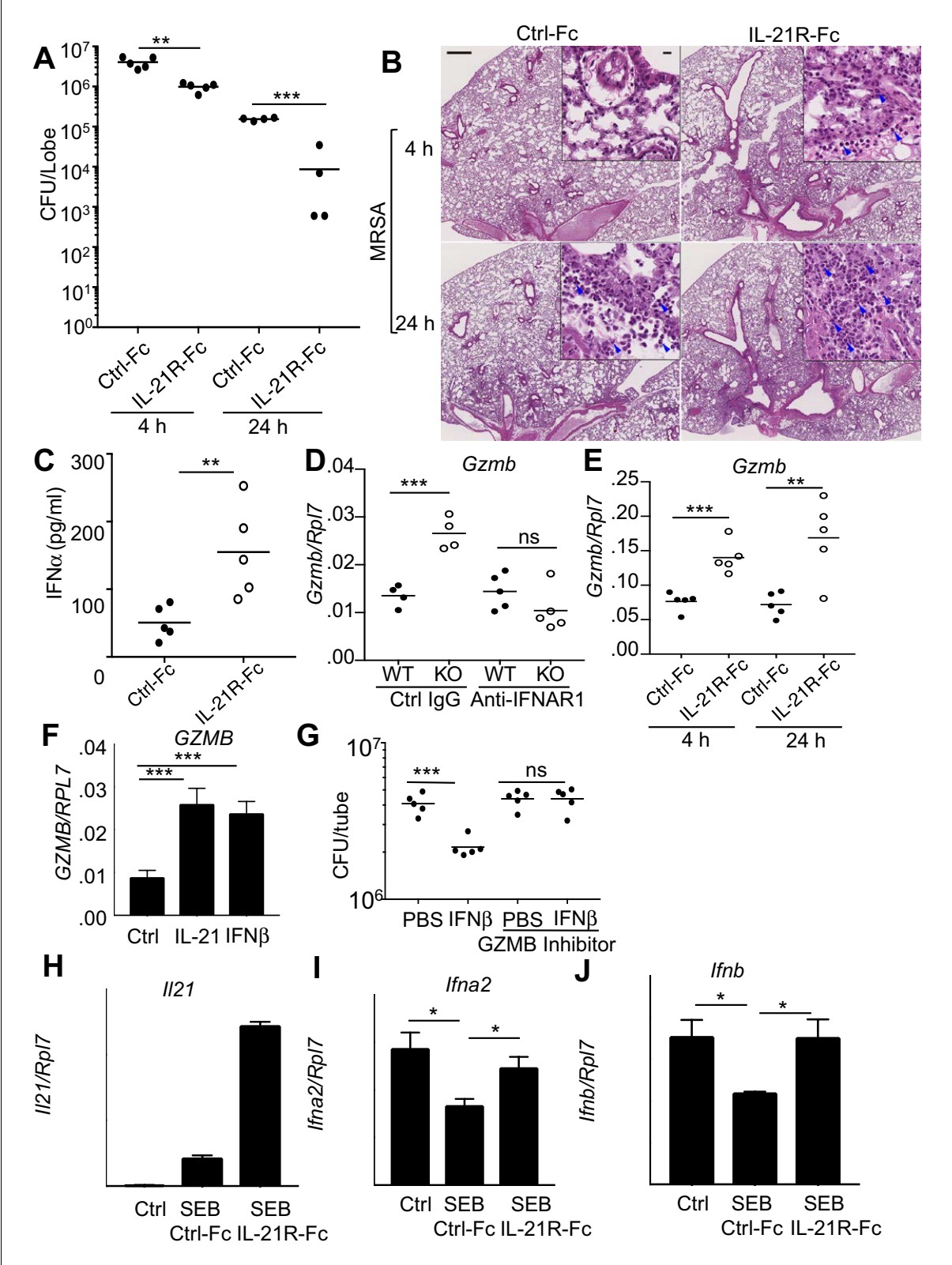

**Figure 6.** Blocking IL-21 signaling with IL-21R-Fc in WT mice phenocopies the enhanced MRSA killing seen in *Il21r* KO mice. (**A**) WT mice were treated intratracheally with 50 µg of control Fc (from IgG1) or IL-21R-Fc protein for 2 days prior to infection with MRSA and lung CFU were quantitated at 4 and 24 hr post-infection with MRSA. (**B**) Histology of lungs in control Fc or IL-21R-Fc pre-treated mice at 4 and 24 hr post-infection (bar in upper left panel = 500 µm; bar in inset = 10 µm). (**C**) IFNα levels in BAL fluid were measured by ELISA 4 hr after infection in mice pretreated with control Fc or IL-
*Figure 6 continued on next page*

Figure 6 continued

21R-Fc. (D) *Gzmb* mRNA was measured in lungs of WT or *Il21r* KO mice pre-treated with isotype control or anti-IFNAR1 antibodies prior to MRSA infection, and normalized to *Rpl7* mRNA expression. (E) *Gzmb* mRNA was measured in lungs of mice treated with either control Fc or IL-21R-Fc prior to MRSA infection and normalized to *Rpl7* expression. (F) Like IL-21, IFNβ also induced *GZMB* mRNA in human peripheral blood neutrophils and normalized to *RPL7* expression. (G) IFNβ induces increased in vitro killing of MRSA by human peripheral blood neutrophils and this was prevented by a granzyme B inhibitor, Z-AAD-CMK. (H–J) Co-cultures of CD4+ T cells and dendritic cells were stimulated with SEB either in the presence of control Fc or IL-21R-Fc, and mRNA was quantitated by RT-PCR after 24 hr and normalized to *Rpl7* expression. Representative experiments are shown; each experiment was performed three times with similar results.

DOI: https://doi.org/10.7554/eLife.45501.012

The following figure supplement is available for figure 6:

**Figure supplement 1.** Pathology and RNA-Seq analysis of lungs from mice treated with control Fc or IL-21R-Fc prior to i.t.

DOI: https://doi.org/10.7554/eLife.45501.013

ND2 (see 'PBS' lanes in *Figure 7A*, middle panel). Interestingly, all six patients had much lower *GZMB* mRNA levels, including in response to IL-21, as compared to the three normal donors (*Figure 7B*). Whereas the basis for this lower basal *GZMB* expression in the AD-HIES patients is unclear, these results indicate that STAT3 deficiency diminished IL-21-mediated enhancement of *GZMB* expression.

In light of the enhanced interferon responses in mice lacking IL-21R signaling, we next examined whether purified neutrophils from AD-HIES patients had altered interferon responses. RNA-Seq analysis was performed using either normal donor or AD-HIES neutrophils stimulated for 4 hr with IL-21, IFNβ, or both cytokines (*Figure 7C* and *Supplementary file 5*). Interestingly, AD-HIES patient neutrophils had significantly enhanced expression of a cluster of IFNβ-responsive genes, as compared to normal donor neutrophils (*Figure 7C*), and we confirmed by RT-PCR that *GBP1* and *GBP2* were induced in neutrophils from additional donors as well (*Figure 7D*). These observations are consistent with higher expression of IFN-related genes in *Il21r* KO than in WT mice (*Figure 5B*).

A mouse model of AD-HIES with a *Stat3* mutant transgene corresponding to a mutation found in a patient with AD-HIES has been shown to mimic the characteristics of AD-HIES patients (*Steward-Tharp et al., 2014*). Accordingly, we treated these transgenic mice with IL-21 prior to i.t. MRSA infection and then analyzed the mice 7 hr after infection. Analogous to our observations of defective MRSA killing by highly purified neutrophils from AD-HIES patients examined in vitro (*Figure 7A*), IL-21 did not enhance MRSA killing in these *Stat3* mutant transgenic mice (*Figure 7E*). Interestingly, however, these mice had enhanced basal killing of pulmonary MRSA in the absence of IL-21 as compared to the corresponding WT mice (*Figure 7E*, compare the PBS-treated samples). Consistent with an inverse relationship between IL-21 signaling and IFNα levels following infection with MRSA, IFNα levels were elevated in both serum (*Figure 7F*) and BAL fluid of the *Stat3* mutant mice (*Figure 7G*). Although there presumably are effects of STAT3 deficiency in addition to the IFN-related connection, our results collectively indicate that the absence of a functional STAT3-mediated immune response correlates with an amplified type I interferon response and that this contributes to the augmented killing observed in this setting (*Figure 8*).

## Discussion

In this study, we demonstrate that IL-21 can act directly on neutrophils and promote an anti-microbial program that leads to the clearance of pulmonary bacterial pathogens. Interestingly, IL-21 augments expression of the IL-21R on neutrophils and additionally upregulates expression of granzyme A, granzyme B, and perforin in enriched human peripheral blood neutrophils and mouse lung neutrophils, with enhanced cytotoxic activity against MRSA and *S. pyogenes*. In human cells, granzymes A and B are delivered to target cells by granulysin, which we show is also induced by IL-21 in human neutrophils, and these enzymes can kill bacteria through cleavage of enzymes that promote oxidative damage, thereby preventing the oxidative stress response (*Walch et al., 2014*). Interestingly, granzyme B was also reported to contain a peptide sequence with bactericidal activity against *S. aureus* (*Shafer et al., 1991*). The idea that granzymes are expressed by neutrophils has been controversial (*Grossman and Ley, 2004*; *Hochegger et al., 2004*; *Metkar and Froelich, 2004*), with some studies not identifying these proteins and others collectively detecting their expression in

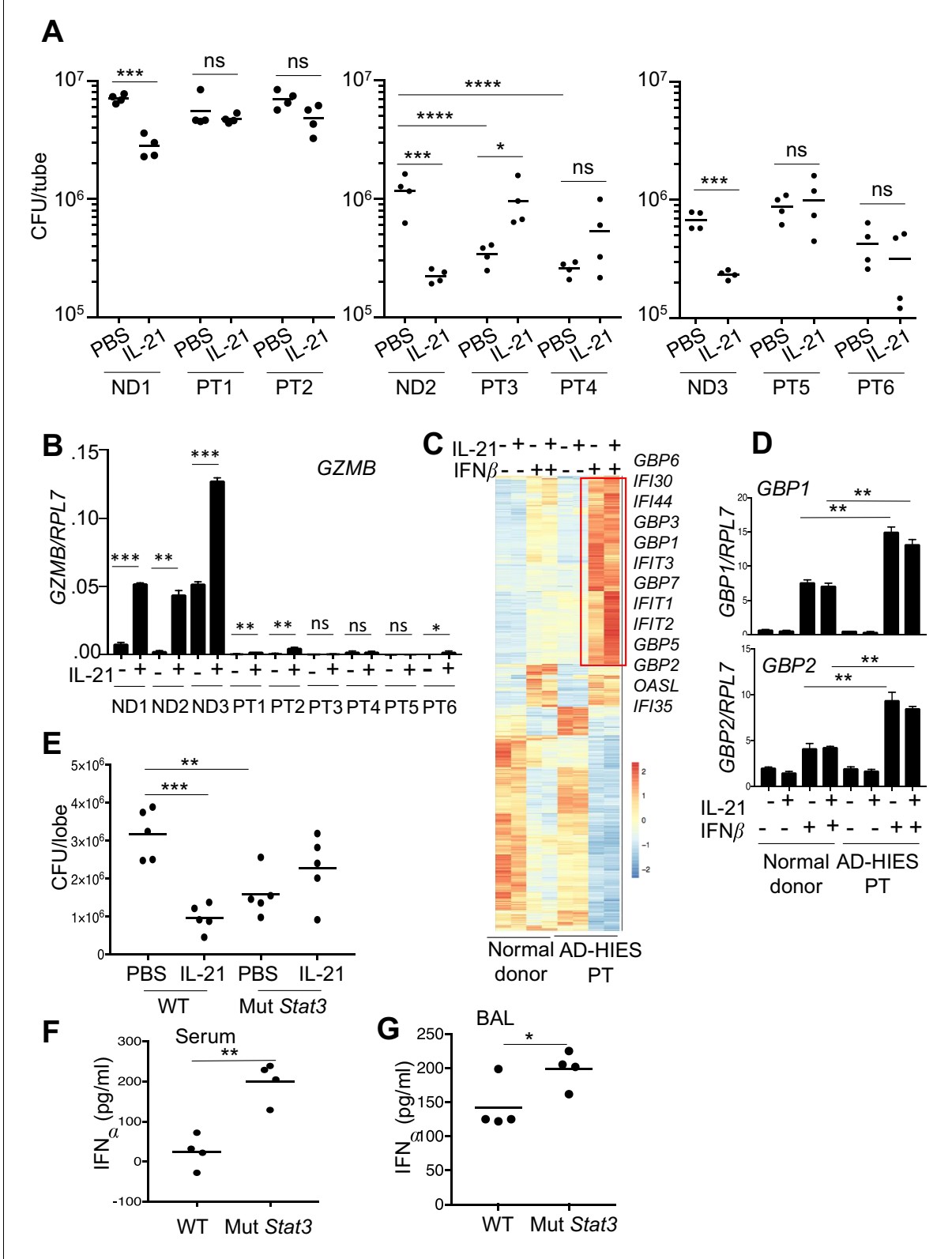

**Figure 7.** Neutrophils from patients with autosomal dominant hyper-IgE syndrome (AD-HIES) display reduced ex vivo IL-21-induced MRSA cytotoxic function. (A) Peripheral blood neutrophils from normal donors (NDs) or AD-HIES patients (PTs) were assayed using an in vitro MRSA killing assay in the absence or presence of IL-21 (100 ng/ml). (B) RT-PCR was used to quantitate *GZMB* mRNA in neutrophils 4 hr after in vitro stimulation without or with IL-21. Expression was normalized to *RPL7* expression (C) RNA-Seq was performed on normal donor and AD-HIES patient neutrophils stimulated for 4 hr

*Figure 7 continued on next page*

*Figure 7 continued*

with PBS, IL-21, IFNβ, or IL-21 +IFNβ. Genes differentially expressed (fold-change >1.5) in two independent RNA-Seq analyses are shown. (D) RT-PCR normalized to *RPL7* expression was used to validate the expression pattern of *GBP1* and *GBP2* in neutrophils from additional normal donors and AD-HIES patients. (E–G) Mutant *Stat3* transgenic mice were treated i.t. with PBS or 2 µg IL-21, infected i.t. 24 hr later with MRSA, and at 7 hr post-infection lung MRSA CFU quantitated (E), and IFNα levels were measured in the serum (F) and BAL fluid (G).

DOI: https://doi.org/10.7554/eLife.45501.014

neutrophils by confocal microscopy, western blotting, and ELISA, and subcellular fractionation has localized granzymes to primary neutrophil granules (*Hochegger et al., 2007*; *Wagner et al., 2004*; *Wagner et al., 2008*). Moreover, our data indicate that intracellular granzyme B is expressed by neutrophils, that its release is induced by IL-21, that depletion of neutrophils decreases IL-21-mediated induction of *Gzmb* mRNA in the lung thereby implicating these cells as a source of these transcripts, and that a granzyme inhibitor blocks the IL-21-induced lowering of MRSA CFU. Interestingly, granzyme B-expressing neutrophils were detected in lung granulomas from humans and macaques in association with *M. tuberculosis* infection (*Mattila et al., 2015*). IL-21 has also been shown to be critical in the adaptive immune response to *M. tuberculosis* (*Booty et al., 2016*), suggesting a multi-faceted IL-21-mediated response to pulmonary microbial infection. Our observation that granzyme B was released by neutrophils after IL-21 stimulation in vitro is consistent with reports of a neutrophil-mediated extracellular proteolytic activity on either extracellular matrix components (*Boivin et al., 2009*) or components of the innate complement system (*Perl et al., 2012*) that may more broadly regulate disease pathology. Human B cells have also been shown to release granzyme B in response to co-stimulation by the B cell receptor and IL-21 (*Hagn et al., 2009*). Thus, IL-21 and granzymes appear to play roles in early responses to pathogens by both innate and adaptive immune cells. Cleavage of IL-1α by secreted granzymes has been shown to enhance the biological activity of this cytokine both in vitro and in vivo, thus adding to the potential functions for secreted granzymes (*Afonina et al., 2011*).

Although IL-21 augmented MRSA killing, to our surprise both *Il21r* KO mice and WT mice treated with an IL-21R-Fc fusion protein also displayed higher pulmonary MRSA clearance than untreated WT mice. This conundrum is at least in part explained by our demonstrating that chronic absence of IL-21 signaling (*Il21r* KO) or even short-term inhibition of IL-21 signaling (treatment with IL-21R-Fc fusion protein) is accompanied by an enhanced type I interferon response in the lung, with increased granzyme B expression and neutrophil anti-microbial action. Although type I IFNs can regulate CD8 cytolytic activity via the induction of granzyme B (*Kohlmeier et al., 2010*) and as noted above, granzyme B-expressing neutrophils were reported in lung granulomas following infection with *M. tuberculosis* (*Mattila et al., 2015*), to our knowledge an interplay between type I IFNs and granzymes in neutrophil anti-microbial action has not been reported. The effects of type I IFN on bacterial infection can be either beneficial or detrimental, depending on the particular bacterium and the context of potential co-infection with a virus (*Stifter and Feng, 2015*; *Trinchieri, 2010*). Here, we show that type I IFN promoted killing of MRSA and that this was inhibited by a granzyme B inhibitor. Previous studies have shown that *S. aureus* induces a type I IFN signal in dendritic cells via TLR9 (*Parker and Prince, 2012b*), and that *Ifnar1* KO mice were protected from MRSA infection compared to WT mice and preferentially survived a lethal dose of MRSA. Our results demonstrate that type I IFN induction of granzyme B can mediate MRSA clearance, and experiments using anti-IFNAR1 in WT mice reduced clearance of MRSA. This implies a potential difference between mice developmentally deficient in expression of receptor for type I IFNs versus acute blockade of IFN signaling. In addition to the effects of type I IFN on MRSA clearance, our RNA-Seq analysis identified type III IFN receptor (IL-28R) as one of the upstream regulators of the enhanced killing by *Il21r* KO mice. Interestingly, type III IFN receptor KO mice were also resistant to MRSA, displaying reduced inflammation and lower expression of inflammatory cytokines and yet no reduction in recruitment of neutrophils to the lung (*Cohen and Prince, 2013*). Some of these effects of type III IFN are mediated by deficient neutrophil serine protease cleavage of IL-1β precursor (*Pires and Parker, 2018*). Thus, compensatory changes in cytokines or cell populations in both type I and type III IFN receptor KO mice may account for differences seen with mice in which IFN is acutely blocked by anti-IFNAR1.

IL-21 signals mainly via STAT3 (*Kwon et al., 2009*; *Zeng et al., 2007*), often in the context of large transcription factor complexes comprising AP1 and IRF4 (*Ciofani et al., 2012*;

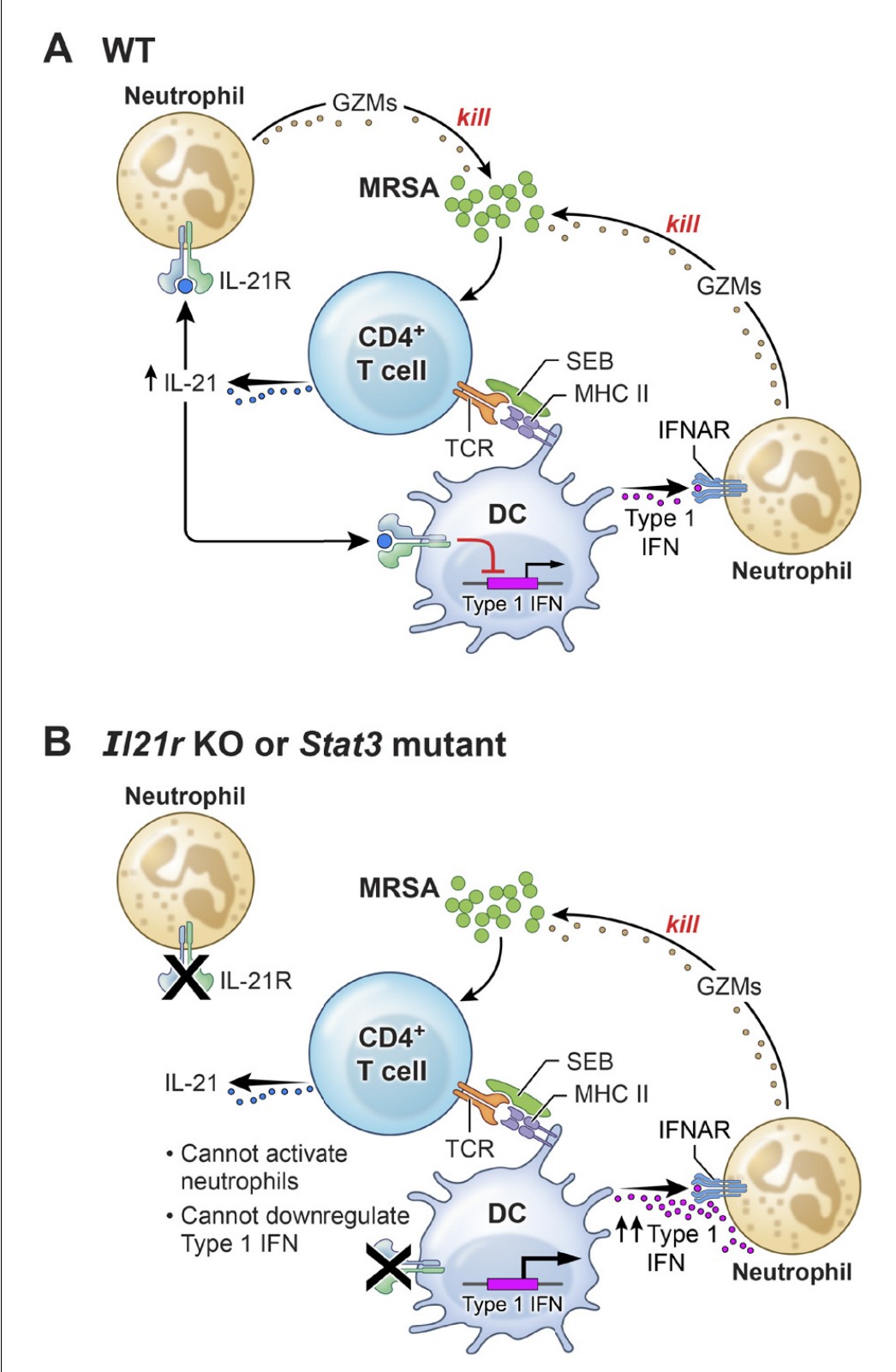

**Figure 8.** A model for the functional interplay of IL-21 with type I interferons in the response to MRSA. (**A**) In wild-type (WT) mice, MRSA produces SEB, which bridges between T cells and dendritic cells by interacting with TCR and MHCII, leading to the production of IL-21 by CD4+ T cells. IL-21 stimulates neutrophils to release granzymes, with enhanced killing of MRSA. Type I IFN (e.g., produced by dendritic cells, as shown in the cartoon) can also induce granzyme production to promote killing of MRSA, but IL-21 inhibits dendritic cell production of type I IFN. (**B**) In the absence of either IL-

*Figure 8 continued on next page*

*Figure 8 continued*

21R or a functional STAT3 signaling response, the IL-21-induced neutrophil granzyme production does not occur. However, IL-21-mediated repression of type I IFN production by dendritic cells also no longer occurs, which leads to enhanced production of type I IFN and hence increased killing of MRSA through this pathway. The functional cross-talk of and relative potency of the IL-21- and type 1 IFN-mediated pathways and levels of each cytokine influence the outcome.

DOI: https://doi.org/10.7554/eLife.45501.015

*Glasmacher et al., 2012*; *Li et al., 2012*); it also more weakly can activate and signal via STAT1 (*Wan et al., 2015*; *Zeng et al., 2007*), with sometimes opposing effects of STAT1 and STAT3 (62). Type I interferons dominantly activate STAT1/STAT2/IRF9 (ISGF3) complexes; thus, it is possible that type I interferons and IL-21 might functionally compete given the enhanced expression of type I IFN-regulated transcripts in *Stat3* KO T follicular helper cells (*Ray et al., 2014*). Moreover, STAT3 has been shown to negatively regulate type I IFN-mediated anti-viral responses (*Wang et al., 2011b*). Importantly, we showed that IL-21-induced granzyme B responses were defective in cells from STAT3-deficient AD-HIES patients, which could relate to the recurrent infections that occur in these individuals, including with *S. aureus* (*Freeman and Holland, 2008*; *Holland et al., 2007*), and furthermore, AD-HIES neutrophils had defective IL-21-induced MRSA cytotoxic activity in vitro. The failure of AD-HIES patients to produce a functional Th17 response is consistent with studies demonstrating the defective clearance of *S. aureus* from the lungs of *Il17r* or *Il22* KO mice (*Kudva et al., 2011*), and this may to be dependent on reduced anti-microbial function of lung epithelial cells (*Minegishi et al., 2009*). However, although IL-21 can be produced by Th17 effector cells, IL-21 production in the lung could be dominated by T follicular helper cells, independent of local production of IL-17, and in fact, IL-17 production can be independent of STAT3 (*St Leger et al., 2018*). Interestingly, neutrophils from some AD-HIES patients had enhanced IL-21-independent in vitro MRSA cytotoxic activity, which could potentially relate to the augmented STAT1 responses that have been observed in CD4$^+$ T cells (*Wan et al., 2015*) and neutrophils (*Holland et al., 2007*) from these patients. However, not all of the AD-HIES patients' neutrophils had enhanced in vitro MRSA cytotoxic activity, potentially reflecting patient heterogeneity and differences in recent infections/medical history. However, when we used a mutant *Stat3* transgenic mouse model for AD-HIES (*Steward-Tharp et al., 2014*), we uniformly found increased levels of type I IFN in both serum and BAL fluid after MRSA infection, suggesting a possible mechanism for the enhanced MRSA killing/lower CFU observed with neutrophils from some AD-HIES patients as well as from the AD-HIES-like *Stat3* mutant transgenic mice. It is also conceivable, however, that the compensatory increase in the type I IFN program observed in the absence of IL-21 signaling or STAT3 expression may involve IL-21-mediated inhibition of type I IFN expression by dendritic cells, as we observed in our in vitro experiments with SEB. It is important to recognize that multiple cytokines besides IL-21 can also activate STAT3, and some of the STAT3-mediated effects on MRSA clearance are known to depend on signaling by IL-6 in the lung epithelial compartment (*Choi et al., 2013*). Nevertheless, our results with *Il21r* KO mice and IL-21R-Fc treated WT mice indicate that the specific absence of IL-21-induced STAT3 signaling can lead to a compensatory interferon-mediated cytotoxic program.

IL-21 can have both pro-inflammatory or anti-inflammatory functional roles, depending on the cellular context and cytokine milieu (*Spolski and Leonard, 2014*), but our current findings importantly extend the known actions of IL-21, particularly related to neutrophil biology and bacterial infection. Our data suggest that IL-21 may not only serve a protective role during pulmonary bacterial infection but also indicate potentially complex homeostatic regulation of the IL-21 versus the type I IFN antimicrobial program, given that the latter is associated with blocking IL-21. These studies underscore the potential complexity of the role of IL-21 in host defense, and further studies are needed to understand how the effects of IL-21 related to neutrophils can integrate with actions of IL-21 on other immune cells. Given that both administering IL-21 (for cancer) and blocking IL-21 (for autoimmune disease) are of potential clinical interest in humans (*Spolski and Leonard, 2014*), it is important to be cognizant of possible effects on host defense of modulating levels and the actions of this cytokine.

# Materials and methods

## Key resources table

| Reagent type (species) or resource | Designation | Source or reference | Identifiers | Additional information |
|---|---|---|---|---|
| Strain, strain background (S. aureus) | FPR3757 | ATCC | ATCC BAA-1556 | |
| Strain, strain background (Streptococcus pyogenes) | NZ131 | ATCC | ATCC-BAA-1633 | |
| Genetic reagent (M. musculus) | Il21r-/- | Ozaki et al. Science 298:1630, 2002 | | |
| Genetic reagent (M. musculus) | mCherry Il21 | Wang et al. P.N.A.S. 108:9542, 2011 | | |
| Genetic reagent (M. musculus) | C57BL/6-Tg(Stat3*)9199Alau/J | Jackson Laboratory | Stock # 027952 | |
| Biological sample (S. aureus) | Staph enterotoxin B | List Biological Laboratories | #122 | |
| Biological sample (S. aureus) | Heat-killed S. aureus | In Vivo Gen | tlrl-hksa | |
| Antibody | Mouse monoclonal anti-granzyme B | Biolegend | GB11 RRID:AB_2562195 | flow cytometry 1:100 |
| Antibody | Mouse monoclonal anti-granzyme B | Biolegend | clone M3304B06 RRID:AB_2565266 | WB (1:500) |
| Antibody | Mouse monoclonal anti-Ly6G (1A8) | BioX Cell | RRID: AB_1107721 | Blocking Ab |
| Antibody | Mouse monoclonal anti-IFNAR (MAR1-5A3) | BioX Cell | RRID: AB_2687723 | Blocking Ab |
| Antibody | Mouse monoclonal MOPC-21 | BioX Cell | RRID: AB_1107784 | Blocking Ab |
| Antibody | Rat monoclonal 2A3 | BioX Cell | RRID: AB_1107769 | Blocking Ab |
| Antibody | Mouse monoclonal anti-human IL21 | BD Biosciences | clone 3A3-N2.1 RRID:AB_2738933 | flow cytometry 1:100 |
| Antibody | Rat monoclonal anti-mouse IL-21R | BD Biosciences | clone 4A9 | flow cytometry 1:100 |
| Antibody | Mouse monoclonal anti-human IL-21R | Biolegend | clone 2G1-K12 RRID:AB_2123988 | flow cytometry 1:100 |

*Continued on next page*

*Continued*

| Reagent type (species) or resource | Designation | Source or reference | Identifiers | Additional information |
|---|---|---|---|---|
| Antibody | Mouse monoclonal anti-human CD66b | Biolegend | clone G10F5 RRID:AB_2563294 | flow cytometry 1:100 |
| Antibody | Mouse monoclonal anti-CD11b | Biolegend | clone M1/70 RRID:AB_312789 | flow cytometry 1:100 |
| Peptide, recombinant protein | Recombinant mouse IL-21 | R and D Systems | 594 ML-100/CF | |
| Peptide, recombinant protein | Recombinant human IL-21 | R and D Systems | 8879-IL-050/CF | |
| Peptide, recombinant protein | Recombinant human GM-CSF | R and D Systems | 215 GM | |
| Peptide, recombinant protein | Recombinant human IL-4 | R and D Systems | 204-IL | |
| Peptide, recombinant protein | Z-AAD-CMK | Enzo Life Sciences | BML-P165 | |
| Commercial assay or kit | IL-21 ELISA | R and D Systems | DY594 | |
| Commercial assay or kit | Granzyme B ELISA | Ebioscience | BMS2027 RRID:AB_2575322 | |
| Commercial assay or kit | IFNalpha ELISA | Ebioscience | 50-246-672 | |
| Commercial assay or kit | Cell Rox Deep Red staining kit | InVitrogen/ Molecular Probes | C10491 | |
| Commercial assay or kit | Mouse neutrophil isolation kit | Miltenyi | 130-097-658 | |
| Commercial assay or kit | Human neutrophil isolation kit | Stem Cell Technologies | 19666 | |
| Commercial assay or kit | KAPA stranded mRNA-Seq library kit | Kapa Biosystems | KK8580 | |
| Chemical compound, drug | Peptidoglycan from S. aureus | Sigma | #77140 | |

## Mice

Sex-matched littermate WT and *Il21r* KO mice were analyzed at 8–12 weeks of age. mCherry-IL-21 reporter mice have been described (*Wang et al., 2011a*). Transgenic mice expressing a mutant *Stat3* that corresponds to mutations found in AD-HIES patients (*Steward-Tharp et al., 2014*) were obtained from the Jackson Laboratory. All experiments were performed under protocols approved by the National Heart, Lung, and Blood Institute Animal Care and Use Committee and followed National Institutes of Health guidelines for use of animals in intramural research.

## Intratracheal inoculation with *S. aureus* and *S. pyogenes*

The USA 300 clinical isolate (FPR3757) of MRSA or the *S. pyogenes* NZ131 strain were used for all experiments. *S. aureus* bacteria were grown overnight at 37°C in a shaking incubator in tryptic soy

broth, diluted 1:10, and mid-logarithmic growth phase bacteria were obtained after an additional 2 hr of culture. *S. pyogenes* were grown in Todd Hewitt Broth under static conditions. Bacteria were pelleted and washed twice with PBS. Bacterial concentrations were estimated by measuring absorbance at 600 nm, and the absolute CFU was determined by plating dilutions on blood agar plates. For infection, mice were lightly anesthetized with ketamine/xylazine and received an intratracheal instillation of 50 μl of *S. aureus* ($2 \times 10^7$ CFU). For neutrophil depletion experiments, mice were injected i.p. with 1 mg 1A8 (anti-Ly6G) mAb or 2A3 isotype control mAb 2 days prior to MRSA infection. For blocking type I interferon responses, mice were injected with 1 mg MAR1-5A3 (anti-IFNAR1) or MOPC-21 isotype control mAbs on each of 2 days prior to infection. All depletion antibodies and isotype controls were from BioXCell. *S. aureus* were quantified after infection by homogenizing one lung lobe in 1 ml PBS, followed by plating serial dilutions on blood agar plates.

## IL-21

We primarily used human and mouse IL-21 from R and D Systems. However, we confirmed key results with IL-21 from Peprotech as well.

## Mouse lung neutrophil isolation and MRSA killing assay

WT mice were treated intratracheally with $2 \times 10^7$ heat-killed *S. aureus* (HKSA, InVivoGen). Lung cells were isolated after one day by digestion with collagenase (1 mg/ml) and DNaseI (1 mg/ml) and were enriched first on a 44%/67% Percoll gradient. Neutrophils were then purified (80–90%) with a mouse neutrophil isolation kit (Miltenyi Biotec) and were used directly in either MRSA killing assays or were in vitro stimulated with IL-21 and then RNA isolated for RT-PCR analysis.

## Mixed lymphocyte cultures

Mouse splenic CD4$^+$ T cells were purified with a Stem Cell Technologies negative selection CD4$^+$ T cell isolation kit and co-cultured with bone marrow-derived dendritic cells (cultured with GM-CSF for 7 days) at 1:1 ratio with or without Staphylococcal entertoxin B (SEB, 0.05 μg/ml, List Biological Laboratories,>95% pure, 14 EU/mg) and in the presence of either IL-21R-Fc blocking agent or control Fc (20 μg/ml) for 24 hr; RNA was then isolated. For intracellular staining of IL-21, human CD4$^+$ T cells were purified from peripheral blood (Dynabeads Untouched negative selection CD4$^+$ T cell kit, Invitrogen) and co-cultured at a 5:1 ratio with monocyte-derived DCs that had been cultured for 6 days in GM-CSF (800 U/ml) plus IL-4 (400 U/ml). At 72 hr, cultures were treated with Golgi Plug for 4 hr and stained for detection of intracellular IL-21 (clone 3A3-N2.1) by flow cytometry.

## Human neutrophil isolation and MRSA killing assay

Whole blood from healthy donors was collected and neutrophils isolated by negative selection using a direct neutrophil isolation kit (Stem Cell Technologies), yielding purity of >99%. For MRSA killing assays, $4 \times 10^5$ neutrophils were incubated in RPMI medium with PBS or IL-21 (100 ng/ml), followed by the addition of 50 μl of MRSA (at a 1:1800 dilution of an optical density at 600 nm ($OD_{600}$) = 0.25) pre-incubated in 10% autologous serum. For granzyme B inhibition, neutrophils were pre-incubated for 20 min with 100 μM Z-AAD-CMK (Enzo Life Sciences). Tubes were rotated at 37°C for 3 hr, and serial dilutions then spread on blood agar plates, incubated overnight, and CFU determined. De-identified whole blood from healthy volunteer donors from the NIH Department of Transfusion Medicine was obtained and met the Office of Human Subjects Research criteria for a waiver for the need for IRB approval.

## Histology

Lungs were inflated before excision, fixed in 10% formalin, embedded in paraffin, and then 5 μm sections were cut and slides were stained with H and E. Clinical scores for pathology and cellular infiltration were independently evaluated by the pathologist.

## Flow cytometry

Whole human blood was incubated at 37°C with either PBS or IL-21 for 3 hr, fixed with warm Phos-Flow Fixative (BD Biosciences), washed twice with FACS buffer and then stained with anti-human IL-21R (2G1-K12) (Biolegend) or isotype control mAbs. For intracellular granzyme B staining, human

neutrophils were purified by sequential LymphoSep and neutrophil purification kit (Stem Cell Technologies), cultured with medium or IL-21 for 4 hr, washed, incubated with human Fc receptor blocking antibodies (BioLegend) for 15 min, and then stained with either CD66b or CD11b with live/dead stain included. Cells were fixed for 30 min, washed with permeabilization buffer (EBioscience FoxP3 staining buffer kit), and stained for 1 hr with PacBlue anti-granzyme B (clone GB11; Biolegend) or isotype control diluted in permeabilization buffer. Bone marrow neutrophils were isolated with negative selection kits (Miltenyi) to >90–95% purity. Mouse bone marrow neutrophils were cultured overnight with IL-21 (R and D Systems) and peptidoglycan (Sigma) prior to IL-21R staining using 4A9 mAb (BD Biosciences). Neutrophils were pretreated with 0.5% acetic acid containing 0.5 M NaCl for 30 s to strip cytokine binding prior to staining with anti-IL-21R antibodies (BD Biosciences). Samples were collected using a BD FACS Canto II and analyzed using Flow Jo software.

## ROS staining of neutrophils
100 μl of cells were suspended in RPMI medium containing 10% FBS and 100 μl (5 μM) of Cell ROX Deep Red was added. Cells were incubated at 37°C for 30 min in medium without or with cytokine, washed twice with PBS, fixed for 15 min at room temperature in 4% paraformaldehyde, and washed twice with FACS buffer. Cells were then surface-stained as usual, washed two times in FACS buffer, and analyzed on a BD FACS Canto II flow cytometer.

## RNA preparation and RNA-Seq
RNA was extracted from lung tissue using Trizol (Invitrogen) and the RNeasy kit (Qiagen). Quantitative RT-PCR was performed using the Omniscript reverse transcription kit, and amplification was performed using the Veriquest PCR master mix and specific Taqman probes from Applied Biosystems. Expression relative to the human *RPL7* or mouse *Rpl7* 'housekeeping' gene was calculated using the Delta Delta Ct method. RNA sequencing was performed using the KAPA Stranded mRNA-Seq Library Preparation kit (Kapa Biosystems) following the manufacturer's instructions using an Illumina HiSeq 2000 or HiSeq 2500 platform (Illumina, San Diego, CA). Sequenced reads (50 bp, single end) were mapped to the mouse genome (NCBI37/mm9, July 2007) using Bowtie 0.12.9, and only the reads that mapped onto exons of each RefSeq gene were measured and normalized using reads per kilobase per million mapped reads. Analyses of differential gene expression were performed using R package edgeR.

## ELISAs
Bronchoalveolar lavage fluid was obtained by flushing the lungs through the trachea with 1 ml PBS. IL-21 was quantitated using an ELISA kit (R and D Systems). Granzyme B was measured using either human or mouse specific ELISA systems (eBioscience), and interferon-α was measured using a mouse specific ELISA that recognizes all subtypes of IFN-α (eBioscience).

## Western blotting for granzyme B
Neutrophils were purified as above, stimulated for 4 hr without or with IL-21 in the presence of protease inhibitor (cOmplete tablets, Roche; one tablet/10 ml). Cells were pelleted at 4°C, washed two times with cold PBS (+PI), and resuspended in Pierce RIPA buffer (+PI). Cells were kept on ice for 30 min with periodic mixing, and extracts were cleared at high speed for 15 min at 4°C. Extracts (10–20 μg/lane) were then run under reducing conditions on an SDS gel (NuPAGE 4–12% Bis-Tris). Western blotting was performed with mouse anti- human granzyme B monoclonal antibody (Biolegend clone M3304B06, final concentration of 1 μg/ml), followed by incubation with goat anti-mouse (IRDye 680RD, Licor) and imaging on an Odyssey CLx.

## Statistical analysis
Two-tailed t tests (nonparametric Mann-Whitney) were performed using Prism (GraphPad). For all statistical analysis. Data were considered significant when $p < 0.05$ (*), $p < 0.01$ (**), $p < 0.001$ (***), or $p < 0.0001$ (****).

## Acknowledgments

This work was supported by the Division of Intramural Research, National Heart, Lung, and Blood Institute and the Division of Intramural Research, National Institute of Allergy and Infectious Diseases. We thank the NHLBI Flow Cytometry core and the NHLBI DNA sequencing core. We thank Dr. Mariana Kaplan, NIAMS and Michail Lionakis, NIAID for critical comments/valuable discussions. MK is supported by a grant from the National Heart, Lung, and Blood Institute (NIH grant 5K22HL125593).

## Additional information

### Funding

| Funder | Grant reference number | Author |
| --- | --- | --- |
| National Institutes of Health | Division of Intramural Research, NHLBI | Warren J Leonard |

The funders had no role in study design, data collection and interpretation, or the decision to submit the work for publication.

### Author contributions

Rosanne Spolski, Conceptualization, Formal analysis, Investigation, Methodology, Writing—original draft, Writing—review and editing; Erin E West, Sharon Veenbergen, Sunny Yung, Majid Kazemian, Jangsuk Oh, Investigation; Peng Li, Zu-Xi Yu, Investigation, Writing—review and editing; Alexandra F Freeman, Stephen M Holland, Resources; Philip M Murphy, Supervision, Writing—review and editing; Warren J Leonard, Supervision, Funding acquisition, Writing—review and editing

### Author ORCIDs

Majid Kazemian (iD) https://orcid.org/0000-0001-7080-8820
Warren J Leonard (iD) http://orcid.org/0000-0002-5740-7448

### Ethics

Human subjects: Blood samples were obtain from normal donors from the NIH Blood Bank under a waiver from the NIH Office of Human Subjects research. Blood samples were also obtained from AD-HIES patients who had given informed consent under an NIH IRB-approved protocol.
Animal experimentation: Experiments involving animals were performed under protocols (H-0087R4) approved by the National Heart, Lung, and Blood Institute Animal Care and Use Committee and followed National Institutes of Health guidelines for use of animals in intramural research.

### Decision letter and Author response

Decision letter https://doi.org/10.7554/eLife.45501.025
Author response https://doi.org/10.7554/eLife.45501.026

## Additional files

### Supplementary files

• Supplementary file 1. RNA-Seq analysis was performed on lung mRNA isolated 7 or 24 hr after treatment of WT mice with PBS or IL-21.
DOI: https://doi.org/10.7554/eLife.45501.016

• Supplementary file 2. RNA-Seq was performed on highly purified human neutrophils incubated for 4 or 24 hr with PBS or IL-21 in the absence or presence of heat-killed *S. aureus*.
DOI: https://doi.org/10.7554/eLife.45501.017

• Supplementary file 3. RNA-Seq analysis from WT and *Il21r* KO mouse lungs 4 and 24 hr after MRSA infection.
DOI: https://doi.org/10.7554/eLife.45501.018

• Supplementary file 4. RNA-Seq was performed on total lung mRNA either before MRSA infection or 4 hr after infection of mice that were pre-treated with either control Fc protein or IL-21R-Fc fusion protein.
DOI: https://doi.org/10.7554/eLife.45501.019

• Supplementary file 5. RNA-Seq analysis of neutrophils from normal human donors and AD-HIES patients; cells were either not treated or treated for 4 hr with IL-21, IFNβ, or IL-21 +IFNβ.
DOI: https://doi.org/10.7554/eLife.45501.020

• Transparent reporting form
DOI: https://doi.org/10.7554/eLife.45501.021

### Data availability

All sequencing data in the final manuscript will be deposited in GEO.

The following dataset was generated:

| Author(s) | Year | Dataset title | Dataset URL | Database and Identifier |
|---|---|---|---|---|
| Spolski R, West EE, Li P, Veenbergen S, Yung S, Kazemian M, Oh J, Yu Z | 2019 | IL-21/type I interferon interplay regulates neutrophil-dependent innate immune responses to Staphylococcus aureus | http://www.ncbi.nlm.nih.gov/geo/query/acc.cgi?acc=GSE129093 | NCBI Gene Expression Omnibus, GSE129093 |

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
