## [Decision Letter]

Thank you for submitting your article "IL-21/type I interferon interplay regulates neutrophil-dependent innate immune responses to *Staphylococcus aureus*" for consideration by eLife. Your article has been reviewed by three peer reviewers, one of whom is a member of our Board of Reviewing Editors, and the evaluation has been overseen by Wendy Garrett as the Senior Editor. The following individual involved in review of your submission has agreed to reveal their identity: Philip Verhoef (Reviewer #2).

The manuscript was reviewed by three experts, including a member of the Board of Reviewing Editors. Overall, the reviewers thought the manuscript was of high interest. Also, reviewers found it of interest and fascinating that the paper described discordant results when examining the role of IL21 in controlling intra-tracheal infections with methicillin-resistant Staphylococcus aureus (MRSA). Administration of IL21 led to enhanced clearance of MRSA, but surprisingly the same result was seen in IL21R ko mice. The authors provide evidence that IL21R ko mice have enhanced type I interferon responses that in part explain the findings.

However, the reviewers were concerned about several issues that are described in detail in the full text of their reviews. Among the main issues raised that require attention are the statistical evaluation of some of the experiments and the need to cite prior work on interferons and MRSA pathogenesis. We invite the submission of a revised manuscript that addresses these issues in particular. The revised manuscript should also address the other concerns listed in the detailed reviews, though no additional new experiments are required for these other concerns.

*Reviewer #1:*

Spolski et al submitted an interesting paper examining the role of IL21 in controlling intra-tracheal infections with methicillin-resistant Staphylococcus aureus (MRSA). They found intra-tracheal administration of IL21 led to enhanced clearance of MRSA, though inexplicably this was relatively small (about 2-fold) in Figure 1 when compared to Figure 2 which demonstrates over a log difference in CFU. They ascribe this effect to neutrophils, but neutrophil depletion led to decreased clearance, regardless of IL21 administration, and the effect of IL21 administration was lost. Most interestingly, MRSA infection of IL21R ko mice also showed enhanced bacterial clearance in a direct comparison experiment. The authors ascribe this to type I interferons which are elevated in infected IL21Rko mice which also show enhanced inflammation histologically. Anti-IFNAR1 resulted in decreased clearance and loss of a difference between WT and IL21R ko mice. Corroborating data are presented supporting the general notion that IL21R ko mice have enhanced type I interferon responses.

There are other experiments that could be done to better describe the effects, such as different doses of MRSA, different assay time points after infection, direct ascertainment of IL21 effects on type I interferon responses. However, I think the paper describes the surprising dual effects of IL21 in bacterial clearance with a plausible explanation of how IL21R ko might lead to enhanced bacterial clearance via type I interferons.

*Reviewer #2:*

Overall, this is a well-done study which demonstrates a novel interplay between IL-21 and type 1 IFN signaling in the context of host defense to Staph aureus infection, with correlates in both mice and humans.

It's a fascinating study in part because the use of the knockout mouse gave a result completely unexpected, but also because the authors explored the mechanism for bacterial clearance in the absence of a functional IL-21 axis and found that type 1 interferon signaling plays a critical role. This raises numerous additional questions, but those need not be addressed in this manuscript. The methodology is thorough and appropriate and the data are appropriately presented. I have no major concerns that would merit significant additional experimentation and think that this manuscript merits publication. I do have several minor concerns that could be addressed (in the minor concerns section).

Minor Comments:

1) Their Introduction provides adequate information about IL-21 and MRSA; however, there was nearly no justification for the exploration of IL-21 as a component of MRSA host-defense. In addition, while I realize that the discovery of type 1 interferons in the absence of IL-21 signals was unexpected, is there a way to work in any information about type 1 interferon signaling in MRSA infection in to the Introduction? Or the relationship between type 1 IFN signaling and IL-21? (some of this is covered in the Results and Discussion…)

2) The authors pre-treat mice with IL-21 1 day prior to infection; however, what effect does IL-21 have in the lungs in the absence of infection? Does it promote recruitment of neutrophils or macrophages? Does it lead to an inflammatory response? Does it pre-activate the immune response to facilitate bacterial clearance? This is not clear from the data presented.

3) Figure 2C clearly indicates that IL-21 has effects beyond those mediated by neutrophils (especially given that this is a log scale). In addition, the authors indicate that IL-21 induces MCP-1 release, which would attract other inflammatory monocytes. Can the authors comment on the additional inflammatory responses that may be activated above and beyond those dependent on neutrophils?

4) In the supplemental data, the authors suggest that CD4 T cells may be a source of IL-21 during MRSA infection. However, a wide range of other cell types may act as sources (notably ILCs) and the subset of CD4 T cells is also not clear (naïve? NKT? Γ δ?) and in addition, I'd imagine this would be an innate response and not an adaptive one. I'm not sure that these data add to the paper, other than indicating that IL-21 increases in the lung in response to infection (so could be playing a role). The authors could consider leaving it out, since it raises more questions than it answers.

5) Throughout the paper, the authors do not statistically analyze apparent increases that are documented in flow cytometry histograms (e.g. Figure 2A, 3B, 3H). Such analysis is easily done and would be informative. This is important, for instance, because as I look at Figure 2A, I definitely believe that PG has a strong effect inducing the IL-21R… but I don't necessarily buy that IL-21 is actually inducing the IL-21R.

6) Figure 6H, the authors indicate that blocking of IL21R strongly enhances IL-21 transcript expression in CD4T cells, which might suggest a feedback inhibition in T cells by IL-21 (if T cells express IL-21R) or suggests that if DCs express IL-21R that they are inhibiting T cell production of IL-21, or that perhaps type IFNs stimulate IL-21 transcription? Can the authors comment on this?

7) Figure 6D should probably be moved to Figure 5, given that Figure 6 otherwise focuses on the effects of blocking the IL-21R.

8) In Figure 4, how do the authors reconcile a higher path score with lower neutrophil counts? What are all the cells that are present in the lungs if they are not neutrophils?

9) Figure 5F: assume that because they didn't indicate it, that there was no statistically significant difference between the various groups?

10) I think that Supplementary Figure 5 (the model figure) is important enough to be in the main manuscript. This is really helpful for clarifying the mechanism.

*Reviewer #3:*

In this study the authors investigated the role of IL-21 during infection with the important human pathogen, Staphylococcus aureus. Application of IL-21 was found to improve bacterial clearance and it did so via a granzyme mediated killing in neutrophils. An unexpected observation was that treatment of mice with IL-21R-Fc or infection of Il21r KO mice also led to an improvement in bacterial clearance. This too was associated with neutrophil mediated killing via IL-21 and correlated with a type I IFN gene signature. Neutrophils from patients with STAT3 mutation also showed decreased granzyme activity. This is a scientifically well carried out study and utilizes a variety of techniques. Enthusiasm is diminished due to the disconnect between the IL-21 and IL-21R-Fc/Il21r-/- KO phenotypes and how can both activation and elimination of a pathway cause the same result. This mechanism is still not fully understood. Also, given the emphasis on the connection to interferon signaling it is surprising that there is an absence of discussion of previous studies given those studies show the opposite phenotype. Specific comments are provided below for the authors.

Minor Comments:

Many points during the manuscript were quantitative differences noted in the text but values were not specified or the% or fold changes noted.

Bacteria were quantified from individual lobes from the lung. This is always a potentially problematic approach, versus a whole lung, as individual lobes can be differentially infected. The lobes that were taken aren't stated.

To clarify, the data in Figure 1F-K is on naïve mice and not in the context of infection?

In Figure 2A it was shown that purified peptidoglycan can increase IL-21R expression, is the same result observed when bacteria are used?

Subsection “IL-21 induces granzyme-mediated MRSA killing by neutrophils”, Figure 2, mentions neutrophil depleted lungs had significant higher cfu compared to non-depleted lungs however, statistics are supplied to support this statement.

IL-21 can induce apoptosis in DCs, is there a difference in cell death of DCs or other cell types in the IL-21 KO and in IL-21 treated mice?

What was the rationale for using heat killed bacteria in Figure 3 versus live?

Data in 4B somewhat replicates that shown in Figure 1.

The Il21r KO mice should have increased inflammation while having reduced neutrophil numbers. What immune cell is thus contributing to this inflammation?

It is not clear the number of biological replicates used to generate the RNA-seq data. It is mentioned two independent experiments were performed, but how many biological samples were included in each experiment?

The heatmap in Figure 5B is only showing interferon related gene products, how were these genes selected?

Purified neutrophils should show a flow diagram in supplemental confirming purity and giving some identification on what other cell types might also be present.

Was the IFN α ELISA used specific to any particular IFN α or was it reactive to all subtypes?

Did treatment of mice with IL-21 thus diminished type I IFN production?

Were all statistics performed using paired t-tests, even for animal studies that are considered to be non-parametric, not normally distributed.

In the studies using IL-21R-Fc and Il21-/- mice, 4h after infection a spike in type I IFN production was observed, in addition to increased inflammation, consistent with a role of type I IFN in immunopathology (see citations below).

In Figure 6 expression of IL-21 and IFN is examined using staphylococcal superantigen B. This is not necessarily the best control given that SEB is not even produced by the strain of MRSA being used. FPR3757 expresses SEK and SEQ, which have shown to be not involved in pathogenesis in mouse models of pneumonia.

What was the source of the SEB used and its purity, i.e. endotoxin levels-which could easily influence IFN induction?

Figure 7B is lacking statistics.

In Figure 7D, would be expectation be IL-21 in WT donors should it increase GBP expression, if the mechanism of IL-21 mediated clearance is analogous to that observed with Il21r-/- and IL-21R-FC experiments.

With the AD-HIES patient cells they were shown to be IFN responsive, did they also show differential interferon induction? Should this IFN responsiveness be also correlated to an improved killing? The stat3 mutant mice were shown to clear better, which is somewhat at odds with AD-HIES patients that have significant problems with S. aureus infection.

Some emphasis was placed on the influence of interferon signaling on the outcome to infection. However, the data in Figure 5E does not show any statistical differences between WT mice given IgG or anti-IFNAR1.

The Discussion doesn't place this work in the context of any of the prior studies that examine MRSA and interferon signaling. Specifically, all the prior studies have shown that both type I and III interferon signaling contributes to pathogenesis. This oversight should be corrected and the contrary results justified. MSSA type I IFN (J Immunol 2012 189:4040, PLoS Path 2014 10::e1003951), III IFN (Eur J Imm 2018 48:1707, PLoS Path 2013 9: e1003682) and in the context of influenza (J Immun 2011 186: 1666, Am J Phys Lung cell Mol 2015 309: L158).

---

## [Author Response]

Reviewer #1:

[…] There are other experiments that could be done to better describe the effects, such as different doses of MRSA, different assay time points after infection, direct ascertainment of IL21 effects on type I interferon responses. However, I think the paper describes the surprising dual effects of IL21 in bacterial clearance with a plausible explanation of how IL21R ko might lead to enhanced bacterial clearance via type I interferons.

We thank reviewer 1 for his/her positive assessment of our manuscript. Regarding the difference in magnitude of the effects of IL-21 on MRSA clearance, mice were pre-treated with isotype control Ig in Figure 2C but not in Figure 1C, but we are not sure if this explains the difference. Importantly, however, the trend was in the same direction. We agree that it is most interesting and confirmatory that MRSA infection of Il21r KO mice also showed enhanced bacterial clearance in a direct comparison experiment.

Reviewer #2:

[…] Minor Comments:1) Their Introduction provides adequate information about IL-21 and MRSA; however, there was nearly no justification for the exploration of IL-21 as a component of MRSA host-defense. In addition, while I realize that the discovery of type 1 interferons in the absence of IL-21 signals was unexpected, is there a way to work in any information about type 1 interferon signaling in MRSA infection in to the Introduction? Or the relationship between type 1 IFN signaling and IL-21? (some of this is covered in the Results and Discussion…)

We noted in the Introduction that IL-21 was known to have a role related to viral infections but there was little information about its roles in the innate function. We now mention that the prior report of IL-21R on human neutrophils motivated us to explore anti-bacterial responses to IL-21 and then that we specifically used MRSA. Regarding mentioning type I interferons in the Introduction, the fact is that we were not focused on type I IFN until we obtained the RNA-Seq results and upstream pathway analysis. We think it makes the most sense to discuss this in the Results and Discussion rather than in the Introduction, but we are happy to consider a specific suggestion if you have a good way to include in the Introduction. We have expanded the discussion of the relationship between type 1 IFN signaling and IL-21 in the Discussion.

2) The authors pre-treat mice with IL-21 1 day prior to infection; however, what effect does IL-21 have in the lungs in the absence of infection? Does it promote recruitment of neutrophils or macrophages? Does it lead to an inflammatory response? Does it pre-activate the immune response to facilitate bacterial clearance? This is not clear from the data presented.

As we noted related to Figure 3C, IL-21 induces expression of *CCL2* (encoding MCP-1), which could be involved in recruitment of myeloid cells to the lung. In fact, IL-21 treatment of naïve mice in the absence of infection leads to a minor enhancement of neutrophil recruitment to the lung (now added as Figure 1—figure supplement 2), which could potentially pre-activate the immune response to facilitate bacterial clearance, but we focused on the responses of neutrophils after MRSA infection.

3) Figure 2C clearly indicates that IL-21 has effects beyond those mediated by neutrophils (especially given that this is a log scale). In addition, the authors indicate that IL-21 induces MCP-1 release, which would attract other inflammatory monocytes. Can the authors comment on the additional inflammatory responses that may be activated above and beyond those dependent on neutrophils?

We agree and the literature supports the idea that IL-21 has effects on both dendritic cell (enhanced apoptosis; Wan et al. Immunity 38:514, 2013) and macrophage populations (enhanced phagocytosis; Vallieres and Girard, 2013). These points and studies are now cited in the manuscript.

4) In the supplemental data, the authors suggest that CD4 T cells may be a source of IL-21 during MRSA infection. However, a wide range of other cell types may act as sources (notably ILCs) and the subset of CD4 T cells is also not clear (naïve? NKT? Γ δ?) and in addition, I'd imagine this would be an innate response and not an adaptive one. I'm not sure that these data add to the paper, other than indicating that IL-21 increases in the lung in response to infection (so could be playing a role). The authors could consider leaving it out, since it raises more questions than it answers.

We previously identified that the IL-21 producing CD4^+^ T cells in the uninfected lung (Spolski et al., 2012) express surface markers consistent with those of T follicular helper cells. Γ δ T cells were ruled out as the source of IL-21 in those studies. We believe that it is important to show that these cells pre-exist in the uninfected lung, consistent with the lung’s representing a barrier immune site, and thus included the data in Figure 1—figure supplement 1. The data could be omitted, although we see no harm in including the results. It is possible that other cell types might contribute.

5) Throughout the paper, the authors do not statistically analyze apparent increases that are documented in flow cytometry histograms (e.g. Figure 2A, 3B, 3H). Such analysis is easily done and would be informative. This is important, for instance, because as I look at Figure 2A, I definitely believe that PG has a strong effect inducing the IL-21R… but I don't necessarily buy that IL-21 is actually inducing the IL-21R.

We have now added in Figure 2—figure supplement 1A panels with MFI values and statistical analysis pertinent to Figure 2A; we also show such data on the right in both Figures 3B and 3H. We agree that the effect of IL-21 on the induction of IL-21R as measured by flow cytometry is small, but it was reproducible and statistically significant. Peptidoglycan clearly has a stronger inducing effect on IL-21R.

6) Figure 6H, the authors indicate that blocking of IL21R strongly enhances IL-21 transcript expression in CD4T cells, which might suggest a feedback inhibition in T cells by IL-21 (if T cells express IL-21R) or suggests that if DCs express IL-21R that they are inhibiting T cell production of IL-21, or that perhaps type IFNs stimulate IL-21 transcription? Can the authors comment on this?

We thank the reviewer for these thoughts. As requested, we now have added the effect of IL-21R-Fc on *Il21* mRNA levels. Although we have not explored the mechanism by which blocking IL21R augments SEB-induced IL21 expression, Strengell et al., 2004, demonstrated that IFN-α in combination with either IL-12 or TCR could enhance expression of IL-21 in human NK and T cells and also downregulate IL-21R expression on these cells.

7) Figure 6D should probably be moved to Figure 5, given that Figure 6 otherwise focuses on the effects of blocking the IL-21R.

We believe that Figure 6 walks the reader through our discovery that IL-21 and type I IFN both induce similar cytotoxic programs to augment MRSA killing. As a result, we think it is logical to present the information in Figure 6 so that they were described in parallel with each other. If you feel strongly, however, we could potentially move Figure 6D to Figure 5.

8) In Figure 4, how do the authors reconcile a higher path score with lower neutrophil counts? What are all the cells that are present in the lungs if they are not neutrophils?

Pathology scores are not only related to neutrophil inflammation. The scores are based on the severity of lung lesions in involved areas, inflammation and alveolitis. WT lung at 24 hours showed only focal lesions, mostly seen at perivascular areas with mild inflammation. These inflammatory cells consist of neutrophils, lymphocytes and macrophages. However, KO lung at 24 hours showed diffuse lesions with alveolitis and numerous neutrophils, lymphocytes and macrophages.

9) Figure 5F: assume that because they didn't indicate it, that there was no statistically significant difference between the various groups?

There was no statistical difference between the groups in Figure 5F. This is now noted in the figure and in the text.

10) I think that Supplementary Figure 5 (the model figure) is important enough to be in the main manuscript. This is really helpful for clarifying the mechanism.

As requested, we have added this figure to the main text as Figure 8.

Reviewer #3:

In this study the authors investigated the role of IL-21 during infection with the important human pathogen, Staphylococcus aureus. Application of IL-21 was found to improve bacterial clearance and it did so via a granzyme mediated killing in neutrophils. An unexpected observation was that treatment of mice with IL-21R-Fc or infection of Il21r KO mice also led to an improvement in bacterial clearance. This too was associated with neutrophil mediated killing via IL-21 and correlated with a type I IFN gene signature. Neutrophils from patients with STAT3 mutation also showed decreased granzyme activity. This is a scientifically well carried out study and utilizes a variety of techniques. Enthusiasm is diminished due to the disconnect between the IL-21 and IL-21R-Fc/Il21r-/- KO phenotypes and how can both activation and elimination of a pathway cause the same result. This mechanism is still not fully understood. Also, given the emphasis on the connection to interferon signaling it is surprising that there is an absence of discussion of previous studies given those studies show the opposite phenotype.

We appreciate that the reviewer thought that this is a “scientifically well carried out study”. We also appreciate that it is hard to reconcile the effects of activation and elimination of the IL21 pathway, but we thought that it was essential to present all of the data together, even if a little confusing. Although the mechanism of this interaction is not understood at the molecular level in neutrophils, we have presented data from 3 different mouse models with IL-21 defective signaling (*Il21r* KO mice, treating WT animals with IL-21R-Fc, and *Stat3* mutant mice) as well as an IL-21 signaling deficient human model (AD-HIES), and these have yielded consistent results with regard to the IL21/type I interferon interaction.

Minor Comments:Many points during the manuscript were quantitative differences noted in the text but values were not specified or the% or fold changes noted.

We have added more quantitative analysis; if additional information is needed, please indicate where you believe it should be done.

Bacteria were quantified from individual lobes from the lung. This is always a potentially problematic approach, versus a whole lung, as individual lobes can be differentially infected. The lobes that were taken aren't stated.

The left single lobe of the lung was chosen for CFU quantitation as this allowed us to both measure bacterial load and assess immune populations or RNA in the remaining right lobes. This is now specified in the legend to Figure 2.

To clarify, the data in Figure 1F-K is on naïve mice and not in the context of infection?

Figure 1F-K are from infected mice. This is now stated in the Figure 1 legend.

In Figure 2A it was shown that purified peptidoglycan can increase IL-21R expression, is the same result observed when bacteria are used?

We used peptidoglycan rather than bacteria so that the longer time point (24hr) could be assessed in the absence of neutrophil death in the cultures.

Subsection “IL-21 induces granzyme-mediated MRSA killing by neutrophils”, Figure 2, mentions neutrophil depleted lungs had significant higher cfu compared to non-depleted lungs however, statistics are supplied to support this statement.

Statistical analysis of this difference has been added to Figure 2C.

IL-21 can induce apoptosis in DCs, is there a difference in cell death of DCs or other cell types in the IL-21 KO and in IL-21 treated mice?

We did not investigate lung DC death in this system; we were concerned that the long enzymatic isolation procedure for purifying lung cells would make it difficult to accurately assess apoptosis in this model. In a previous paper from our lab, Wan et al. (Wan et al., 2013) investigated in vivo death of splenic DCs induced by IL-21 released from α-gal-cer induced NKT cells.

What was the rationale for using heat killed bacteria in Figure 3 versus live?

Heat-killed bacteria were used so that longer (24 h) time points could be included in the RNA-Seq analysis of neutrophil in vitro responses. With live bacteria, the culture would have been overgrown by 24 h. This is now mentioned in the legend to Figure 3.

Data in 4B somewhat replicates that shown in Figure 1.

We agree that Figure 4B is slightly redundant with both Figure 1 and also Figure 4A, but we wanted to assay both the IL-21 treatment as well as the *Il21r* KO samples at the same time points and within a single experiment so that we could directly compare the effects on MRSA clearance. We therefore believe it is best to keep all three panels.

he Il21r KO mice should have increased inflammation while having reduced neutrophil numbers. What immune cell is thus contributing to this inflammation?

As described above and in the text related to Figure 4, the pathology score is not limited to neutrophil numbers and is based on the severity of lesions in affected regions. Lymphocytes and macrophages were also observed.

It is not clear the number of biological replicates used to generate the RNA-seq data. It is mentioned two independent experiments were performed, but how many biological samples were included in each experiment?

RNA-Seq analysis was performed in 2 independent experiments in each of Figure 1G, 3C, 5, and 7C, and in each experiment, 3 to 5 biological samples were pooled for the sequencing reactions. Specific gene expression was validated by RT-PCR or ELISA analysis of additional mice or donors.

The heatmap in Figure 5B is only showing interferon related gene products, how were these genes selected?

Upstream pathway analysis of the RNA-Seq data revealed that IFNAR1 was a top regulator. This led us to focus on the type I interferon pathway.

Purified neutrophils should show a flow diagram in supplemental confirming purity and giving some identification on what other cell types might also be present.

Figure 2—figure supplement 1B shows the purity of mouse lung neutrophils and Figure 3—figure supplement 1A shows the purity of human peripheral blood neutrophils. We do not know the contaminating cells in the mouse lung but based on lack of CD11b expression, they are not macrophages. The human neutrophils are devoid of NK cells or CD8 T cells and are 99.9% pure.

Was the IFN α ELISA used specific to any particular IFN α or was it reactive to all subtypes?

The IFNalpha ELISA used in Figure 5D was reactive to all subtypes. This is now stated in the Materials and methods.

Did treatment of mice with IL-21 thus diminished type I IFN production?

IL-21 did not have an effect on lung type I IFN production (based on RNA-Seq analysis) during the time period of the infection.

Were all statistics performed using paired t-tests, even for animal studies that are considered to be non-parametric, not normally distributed.

All statistics used Mann-Whitney non-parametric t-tests in light of both the small sample size and the animal studies. This is stated in the Materials and methods.

In the studies using IL-21R-Fc and Il21-/- mice, 4h after infection a spike in type I IFN production was observed, in addition to increased inflammation, consistent with a role of type I IFN in immunopathology (see citations below).

We agree and the citations have been incorporated into the Discussion.

In Figure 6 expression of IL-21 and IFN is examined using staphylococcal superantigen B. This is not necessarily the best control given that SEB is not even produced by the strain of MRSA being used. FPR3757 expresses SEK and SEQ, which have shown to be not involved in pathogenesis in mouse models of pneumonia.

We used SEB not as representative of an effect of this MRSA strain but as a convenient superantigen bridge to explore interactions between cells known to produce either IL-21 or type I interferons. SEB helped us to evaluate the importance of the CD4 and DC cooperation for the induction of *Il21, Ifna2*, and *Ifnb* mRNA shown in Figure 6H-J.

What was the source of the SEB used and its purity, i.e. endotoxin levels-which could easily influence IFN induction.

SEB was obtained from List Biological Laboratories and is estimated to be >95% pure with 14 EU/mg. This is now stated in the Materials and methods.

Figure 7 is lacking statistics.

Statistical analysis has now been added to this figure.

n Figure 7D, would be expectation be IL-21 in WT donors should it increase GBP expression, if the mechanism of IL-21 mediated clearance is analogous to that observed with Il21r-/- and IL-21R-FC experiments.

Note that AD-HIES patient neutrophils had greater expression of a cluster of IFNb-responsive genes, as compared to normal donor neutrophils. It is IFNb rather than IL-21 that induces GBP levels in Figure 7D, so we do not expect a greater direct effect of IL-21 on GBP expression. In infected mice in Figure 5D, please note that the KO mice have higher IFNα production and correspondingly higher *Gbp1* mRNA expression in Figure 5G.

With the AD-HIES patient cells they were shown to be IFN responsive, did they also show differential interferon induction? Should this IFN responsiveness be also correlated to an improved killing? The stat3 mutant mice were shown to clear better, which is somewhat at odds with AD-HIES patients that have significant problems with S. aureus infection.

Neutrophils do not produce type IFN; thus, only IFN-responsive genes were identified by RNA-Seq. We did not have access to patient serum samples; therefore, we do not know if other cells in the patients are producing IFN following infection. Although the STAT3 mutant TG mice had enhanced clearance of MRSA in the basal state, IL-21 did not enhance MRSA killing, which was consistent with the inability of IL-21 to induce in vitro MRSA killing by AD-HIES patient neutrophils. In fact, 2 out of 6 of the patient neutrophils had enhanced killing of MRSA in the absence of in vitro stimulation.

Some emphasis was placed on the influence of interferon signaling on the outcome to infection. However, the data in Figure 5E does not show any statistical differences between WT mice given IgG or anti-IFNAR1.

Statistical analysis has been added to Figure 5E, and there is a difference in WT mice given IgG versus anti-IFNAR1.

The Discussion doesn't place this work in the context of any of the prior studies that examine MRSA and interferon signaling. Specifically, all the prior studies have shown that both type I and III interferon signaling contributes to pathogenesis. This oversight should be corrected and the contrary results justified. MSSA type I IFN (J Immunol 2012 189:4040, PLoS Path 2014 10::e1003951), III IFN (Eur J Imm 2018 48:1707, PLoS Path 2013 9: e1003682) and in the context of influenza (J Immun 2011 186: 1666, Am J Phys Lung cell Mol 2015 309: L158).

We thank the reviewer for his/her comments. These previous studies have now been included in our Discussion.